# Two neuropeptides that promote blood feeding in *Anopheles stephensi* mosquitoes

**Prashali Bansal\*, Roshni Pillai[†], Pooja DB, Sonia Q Sen\***

Tata Institute for Genetics and Society, Bangalore, India

## eLife Assessment

This is a **valuable** study that integrates behavioral and molecular approaches to identify neuromodulators influencing blood-feeding behavior in the disease vector *Anopheles stephensi*. Through gene expression analyses across blood-seeking life stages and RNA interference experiments, the authors present **solid** evidence that co-knockdown of the neuromodulators *short neuropeptide F* and *RYamide* affects blood-seeking states in *A. stephensi*. However, evidence demonstrating that these neuropeptides are sufficient to promote host-seeking is lacking.

**\*For correspondence:**
prashali.bansal@tigs.res.in (PB);
sonia.sen@tigs.res.in (SQS)

**Present address:** [†]Institute of Developmental Biology and Neurobiology, Johannes Gutenberg University, Mainz, Germany

**Abstract** Animals routinely need to make decisions about what to eat and when. These decisions are influenced not only by the availability and quality of food but also by the internal state of the animal, which needs to compute and give weights to these different variables before making a choice. Feeding preferences of female mosquitoes exemplify this behavioural plasticity. Both male and female mosquitoes usually feed on carbohydrate-rich sources of nectar or sap but the female also feeds on blood, which is essential for egg development. This blood-appetite is modulated across the female's reproductive cycle, yet little is known about the factors that bring it about. We show that mated, but not virgin *Anopheles stephensi* females, a major vector of urban malaria in the Indian subcontinent and West Africa, suppress blood feeding between a blood meal and oviposition. We identify several candidate genes through transcriptomics of blood-deprived and -sated *An. stephensi* central brains that could modulate this behaviour. We show that *short neuropeptide F (sNPF)* and *RYamide (RYa)* act together to promote blood feeding and identify a cluster of cells in the subesophageal zone that expresses *sNPF* transcripts only in the blood-hungry state. Such females also have more *sNPF* transcripts in their midguts. Based on these data, we propose a model where increased *sNPF* levels in the brain and gut promote a state of blood-hunger, which drives feeding behaviour either by *sNPF*'s action in the two tissues independently or via a communication between them. This occurs in the context of the action of *RYa* in the brain.

## Introduction

An animal's dietary choice is an outcome of its internal physiological state and external stimuli. Its changing nutritional demands, reproductive state, and developmental history are integrated with external cues before a food choice is made. For example, when deprived of a particular nutrient, like proteins, animals show a specific increase in protein-appetite (*Corrales-Carvajal et al., 2016*; *Khan et al., 2021*; *Heeley and Blouet, 2016*). Mating influences food choices of female flies by increasing their preference for proteins, sugars, and salt for egg development and oviposition (*Carvalho et al., 2006*; *Vargas et al., 2010*; *Ribeiro and Dickson, 2010*; *Walker et al., 2015*; *Laturney et al., 2023*).

Changes in both sensory perception and central brain circuits have been shown to underlie such behavioural changes. For example, sensory neurons of starved animals become more sensitive to attractive cues and less sensitive to aversive ones, both in vertebrates and invertebrates (*Inagaki et al., 2012*; *Inagaki et al., 2014*; *Marella et al., 2012*; *Devineni and Scaplen, 2021*; *Toshima and Tanimura, 2012*; *Ganguly et al., 2021*; *Matty et al., 2022*; *Waterson and Horvath, 2015*). These changes are often brought about by neuromodulators and neuropeptides (*Inagaki et al., 2012*; *Inagaki et al., 2014*; *LeDue et al., 2016*; *Ko et al., 2015*; *Root et al., 2011*; *Horio and Liberles, 2021*; *Chen et al., 2015*; *Fu et al., 2019*). In the central brain, internal nutrient or hormone sensors can monitor the animal's nutritional state. In mice, neurons in the hypothalamus express receptors for hunger and satiety signals like ghrelin and leptin (*Waterson and Horvath, 2015*; *Andermann and Lowell, 2017*; *Wee et al., 2024*). In insects, several neurons in the mushroom body, central complex, and subesophageal zone (SEZ) have been shown to integrate hunger and satiety signals to regulate feeding on sugars or proteins, choose between exploration or exploitation of food sources, or initiate or terminate feeding (*Shiu et al., 2022*; *Sayin et al., 2019*; *Tsao et al., 2018*; *Sareen et al., 2021*; *Park et al., 2016*; *Musso et al., 2021*; *Goldschmidt et al., 2023*; *Liu et al., 2017*; *Münch et al., 2022*).

Female mosquitoes offer a striking model for dissecting how internal state and external cues shape dietary choices. Many female mosquitoes have a dual appetite for carbohydrate-rich nectar or sap – to meet energetic requirements – and protein-rich blood – for egg development. Which meal a female chooses to take is influenced by nutritional demands, reproductive state, and developmental history. For example, mosquitoes with low teneral reserves will take multiple blood meals for egg development (*Takken et al., 1998*; *Farjana and Tuno, 2013*; *Scott and Takken, 2012*; *Ramasamy et al., 2000*). Nutritional state also influences other internal state outputs on blood feeding. For example, whether virgin females take blood meals or not is dependent on prior sugar feeding in most *Aedes* strains (*League et al., 2021* and references within).

Despite these species- and context-dependent variations, a consistent finding is that female mosquitoes oscillate between states of heightened and suppressed host seeking across successive reproductive cycles (*Figure 1A*). For example, in *Ae. aegypti* and *An. gambiae*, newly emerged mosquitoes display no or low interest in the host, which increases as they age (*Davis, 1984a*; *Takken et al., 1998*; *Foster and Takken, 2004*; *Ramírez-Sánchez et al., 2023*) and once fed to repletion, host attraction is suppressed. The strength and timing of this suppression varies between species — being robust and well-characterised in mated *Ae. aegypti* (*Klowden and Lea, 1979a*; *Klowden and Lea, 1979b*; *Duvall et al., 2019*), and more flexible in anophelines (*Klowden and Briegel, 1994*; *Takken et al., 2001*; *Scott and Takken, 2012*). Sugar feeding is not modulated in this way, in some cases it is preferred soon after emergence (*Foster, 1995*; *Foster and Takken, 2004*).

The modulation of blood feeding provides a useful framework for understanding how internal and external cues shape meal choice. What underlying molecular cues represent internal physiological states to influence dietary choice is of interest not only because of its importance in public health, but also because such understanding has implications for other organisms. In *Ae. albopictus* anticipatory expression of the yolk protein precursor gene *vitellogenin 2* in the fat body promotes host-seeking behaviour (*Dittmer et al., 2019*). In *Ae. aegypti*, neuropeptide Y-like receptor 7 (NPYLR7), and more recently, short neuropeptide F (sNPF) and neuropeptide RYamide (RYa) have been shown to suppress host attraction, while neuropeptide F (NPF) promotes it (*Liesch et al., 2013*; *Christ et al., 2017*; *Zeledon et al., 2024*; *Dou et al., 2024*; *Duvall et al., 2019*). Factors from the host can also both promote (ATP) *Galun et al., 1963* and terminate feeding (Fibrinopeptide A; *Sakuma et al., 2024*).

Here, we describe the blood- and sugar-feeding patterns of *Anopheles stephensi*, which is endemic to the Indian sub-continent but has expanded its range into West Africa, putting millions more people at risk of malaria (*Sinka et al., 2020*). Neurotranscriptomics across the different conditions of blood-deprivation and -satiety allowed us to shortlist several candidate molecules that might promote blood-feeding behaviour. Through dsRNA-mediated knockdown of nine of them, we identified two neuropeptide genes – *sNPF* and *RYa* – that together promote blood feeding through their action in the brain and possibly the abdomen. We find that *sNPF* expression pattern reflects this requirement: a cluster of cells in the SEZ expresses *sNPF* transcripts only in the blood-hungry females and such females also have more *sNPF* transcripts in their midguts. *RYa* is not modulated similarly. Both receptors are expressed in the brain and the *sNPF receptor* is also expressed in the midgut. Together,

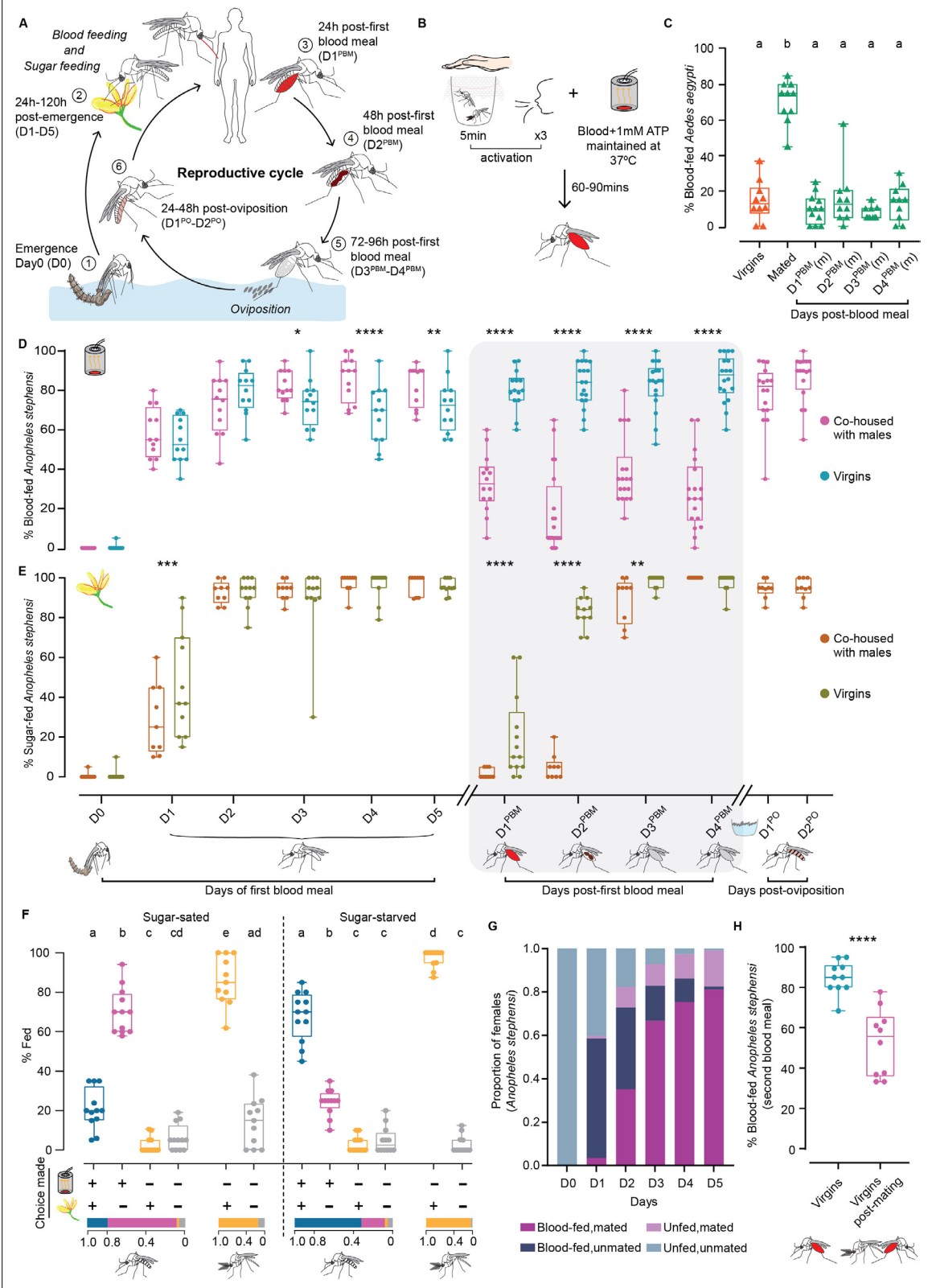

**Figure 1.** Feeding behaviours of *An. stephensi*. (**A**) Reproductive cycle of *An. stephensi*. Upon emergence [1, D0-D5] females have a dual appetite for protein-rich blood and carbohydrate-rich sources of sugar [2]. After a blood meal [3,4,5, D1$^{PBM}$-D4$^{PBM}$], the eggs mature and are laid in water [6]. The next reproductive cycle can now begin. Striped abdomens indicate mated females. (**B**) Blood-feeding assay. 0–7 hr-old mosquitoes were collected in cups and aged appropriately on sugar. Prior to the test, mosquitoes were 'activated' for 5 min by presenting a human hand, followed by three exhalations.

*Figure 1 continued on next page*

*Figure 1 continued*

Blood meals were presented through a Hemotek perfumed with human skin odours. Blood-fed mosquitoes were visually scored. (**C**) Blood-feeding behaviour of *Ae. aegypti*. 8-day-old virgin and mated females were assessed for first blood meal. Fully fed mated females were assessed for second blood meal (D1$^{PBM}$-D4$^{PBM}$). 18–21 females/trial, n=10–12 trials/group. Kruskal-Wallis with Dunn's multiple comparisons test, p<0.05. Data labelled with different letters are significantly different. (m): mated. (**D**). Blood-feeding behaviour of *An. stephensi*. Co-housed and virgin females were tested on the day of emergence (D0) and each day for the next 5 days (D1–D5). D1$^{PBM}$-D4$^{PBM}$ represents blood feeding 1–4 days post-first blood meal and D1$^{PO}$-D2$^{PO}$ represents blood feeding 1–2 days post-oviposition. 18–21 females/trial, n=9–20 trials/group. Generalized Linear Mixed Model with post-hoc pairwise comparisons using estimated marginal means and Bonferroni-corrected p-values; *p<0.05; **p<0.01; ****p<0.0001. (**E**) Sugar-feeding behaviour of *An. stephensi*. Females of similar conditions as in (**D**) were tested for sugar feeding. 18–21 females/trial, n=9–20 trials/group. Generalized Linear Mixed Model with post-hoc pairwise comparisons using estimated marginal means and Bonferroni-corrected p-values; **p<0.01; ***p<0.001; ****p<0.0001. (**F**) Choice assay for blood and sugar. Sugar-sated or sugar-starved females, co-housed with males (indicated by striped abdomens) were given a choice between blood and sugar and assessed for the choice made (bottom): blood and sugar (blue); blood only (magenta); sugar only (orange); none (grey). Co-housed males were assessed for sugar feeding only. Proportion of mosquitoes fed on each particular choice are shown at the bottom. 17–24 mosquitoes/trial; n=11–12 trials/condition. One-way ANOVA with Holm-Šídák multiple comparisons test, p<0.05. Data labelled with different letters are significantly different. (**G**) Mating and blood feeding. Mating status of D0-D5 co-housed females assayed in (**D**) were determined post-hoc via spermatheca dissection. n=232–239 females analysed for each time point. (**H**) Mating after blood feeding. Blood-fed virgins were allowed to mate and assayed for second blood meal. Age-matched virgins were used as controls. 15–20 females/trial, n=10 trials. Unpaired t-test; ****p<0.0001.

The online version of this article includes the following figure supplement(s) for figure 1:

**Figure supplement 1.** Standardisation of various behavioural assays to study feeding behaviours in *An. stephensi*.

these data are consistent with a model where an increase in levels of *sNPF* in the brain and the midgut promotes a state of blood-hunger in *An. stephensi*. This occurs in the context of *RYa* signalling in the brain.

## Results

### Blood feeding, but not sugar feeding, is modulated across the reproductive cycle

To assess the female *An. stephensi* appetite for blood through the reproductive cycle, we designed a behavioural assay (*Figure 1B*, *Figure 1—figure supplement 1A*; see Materials and methods for details), and tested it on *Ae. aegypti*, a well-studied model. We were able to reproduce the previously reported results that mating enhances blood feeding in *Aedes* (*Venkataraman et al., 2023*; *Villarreal et al., 2018*; *Adlakha and Pillai, 1976*) and that once fed to repletion, females actively suppress blood feeding until oviposition (*Duvall et al., 2017*; *Figure 1C*).

We used this assay on *An. stephensi*, and employed Generalised Linear Mixed Model (GLMM) to compare appetites across conditions (see *Supplementary file 1*). We found that these females were uninterested in blood when they emerge but that their appetite increased with age (magenta, D0-D5 in *Figure 1D*; *Supplementary file 1*: model 1). Once fed to repletion, like *Ae. aegypti*, *An. stephensi* also suppress blood feeding for at least four days after a blood meal (magenta, D1$^{PBM}$-D4$^{PBM}$ in *Figure 1D*) and until oviposition, after which, the suppression was lifted to pre-blood meal levels (magenta, D1$^{PO}$-D2$^{PO}$ in *Figure 1D*). Unlike blood feeding, sugar feeding was not modulated similarly barring a brief suppression post-blood meal, which was not dependent on oviposition (*Figure 1E*; methods and *Figure 1—figure supplement 1G* for details, *Supplementary file 1*: models 4, 5, 6).

To ascertain whether sugar and blood represent parallel appetites independent of each other, we presented sugar-sated co-housed females a choice between the two. Under these conditions, 70% of the females chose to feed on blood alone and only 21% took both meals (*Figure 1F*). When co-housed females were deprived of sugars for a day prior to testing, the pattern reversed: 68% of the females now took both sugar and blood meals and only 24% fed on blood alone (*Figure 1F*). As expected, males fed robustly on sugar in both conditions (grey, *Figure 1F*).

Since females took blood meals regardless of their prior sugar feeding status and only sugar feeding was selectively suppressed by prior sugar access, we infer that these two represent parallel appetites regulated independently. The fact that most females in the starved group took both sugar- and blood meals, while most females in the sated condition ignored sugar, supports this interpretation. An alternate interpretation of these data are that two represent hierarchical appetites: that some sugar feeding is necessary to stimulate blood feeding. Determining the order of feeding events in

the choice assay would be necessary to comment on this. However, prior reports that females can survive on blood meals alone (with no access to plant sugars *Scott and Takken, 2012*) argue against this model. Together, these data suggest that in newly emerged females, blood and sugar represent parallel appetites. We note that their interdependence may emerge later in the reproductive cycle (D1$^{PBM}$-D2$^{PBM}$ in *Figure 1E*).

As mating promotes the first blood meal in *Ae. aegypti*, we tested if this was true for *An. stephensi*. A post-hoc dissection of each female's spermatheca (where sperm is stored after mating) revealed that many of the blood-fed females in the above assay were unmated (*Figure 1G*). To confirm this, we assayed blood-feeding behaviour in virgin females and found that virgin *An. stephensi*, unlike *Ae. aegypti*, have a robust appetite for blood and that mating enhances it only marginally (cyan, D0-D5, *Figure 1D*; *Supplementary file 1*: model 1, 2). Interestingly, however, these virgins, unlike mated *An. stephensi* (or *Ae. aegypti*) females, failed to suppress their blood-appetite, even after engorging on blood. Instead, they continued to blood-feed for up to four days after their first blood meal (cyan, D1$^{PBM}$ - D4$^{PBM}$ in *Figure 1D*; *Supplementary file 1*: model 3). A flipped order of the two behaviours – virgins were first given a blood meal then allowed to mate – also resulted in suppression of the appetite (*Figure 1H*). This suggests that the post-blood meal suppression of blood-appetite in *An. stephensi* is dependent upon mating status.

Together, these data describe female *An. stephensi*'s feeding behaviours: (1) in newly emerged females blood and sugar represent parallel appetites, (2) blood but not sugar-appetite (barring a brief effect) is suppressed between a blood meal and oviposition, and (3) mating is not necessary for blood feeding in newly emerged females, but it is necessary for its suppression between a blood meal and oviposition.

## Host seeking is modulated across the reproductive cycle

The modulation of blood feeding through the reproductive cycle could be due to (1) modulation in peripheral sensitivity to host kairomones, (2) modulation by central brain circuits, or (3) modulation in both peripheral and central circuit elements. Extensive data in other mosquito species suggest that sensitivity to host kairomones changes as females age (*Grant and O'Connell, 1996*; *Davis, 1984a*; *Omondi et al., 2019*; *Hill et al., 2021*; *Tallon et al., 2019*) and following a blood meal (*Davis, 1984b*; *Takken et al., 2001*; *Fox et al., 2001*). We tested this in *An. stephensi*.

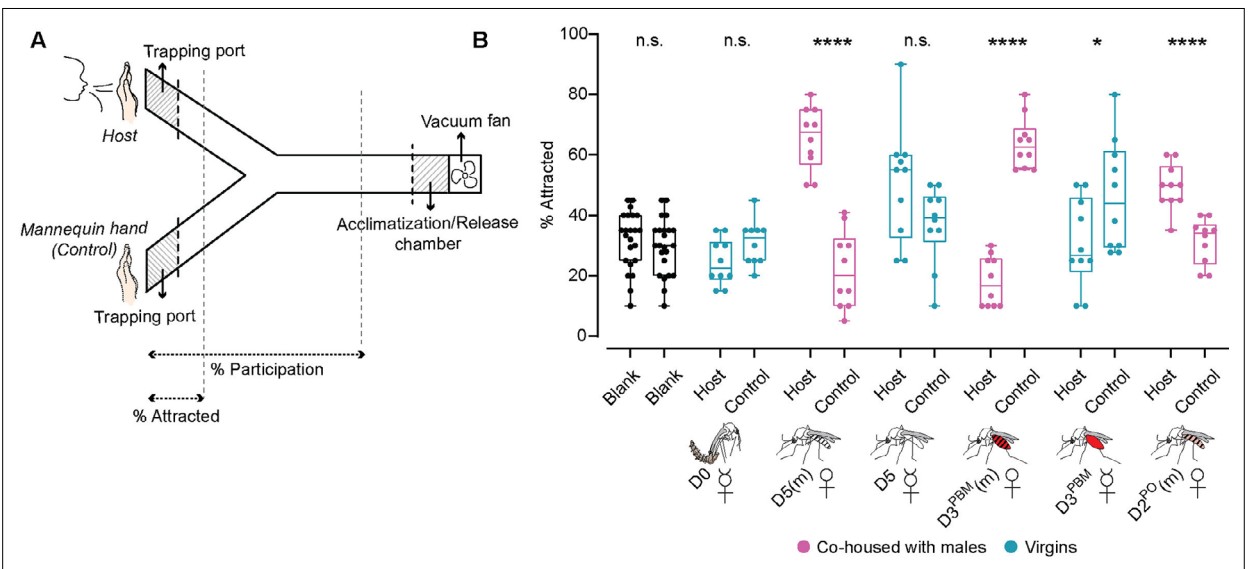

**Figure 2.** Host seeking is modulated like blood feeding is in *An. stephensi*. (**A**) Y-maze schematic for host-seeking behaviour of female *An. stephensi*. Females were acclimatised in the chamber for 5 min while being exposed to host kairomones presented in a test arm. A fan sucked the air from both host and control arms at 0.3–0.6 m/s. Post-acclimatisation, mosquitoes were released and allowed to choose between the two arms. (**B**) Percent females attracted to either host cues or control arm at the indicated time points. 17–21 females/trial, n=10 trials/group. Unpaired t-test; n.s.: not significant, p>0.05.; * p<0.05; ****p<0.0001. (D0: day of emergence; D5(m): 5 days post-emergence, mated; D5: 5 days post-emergence, virgin; D3$^{PBM}$(m): 3 days post-blood meal, mated; D3$^{PBM}$: 3 days post-blood meal, virgin; D2$^{PO}$(m): 2 days post-oviposition, mated).

Because of the multi-sensory nature of host seeking, we used a two-choice behavioural assay where multiple host cues were provided in one arm while the other had none. Mosquitoes had to make a choice between the two arms (*Figure 2A*). When no host cues are provided in either (blank) even blood-hungry females distribute equally between the two arms demonstrating the assay has no bias (black, *Figure 2B*). Using this assay, we found that newly emerged females showed no preference for either the control or the host arm (*Figure 2B*). Five days later, they were strongly attracted to the host cues and this attraction was suppressed after their blood meal and until oviposition (magenta, *Figure 2B*). This general trend in attraction to host cues tracks the trend in blood feeding. Interestingly, the response of virgin females to host cues in the Y-maze was highly variable under all conditions tested (cyan, *Figure 2B*). Yet, like in the blood-feeding assay, blood-fed virgins were comparatively more attracted to human host cues than were the mated blood-fed females (*Figure 2B*).

These data suggest that, like blood feeding, host seeking is also modulated through the female's reproductive cycle, and mating influences this modulation by enhancing it initially and suppressing it post-blood meal. The variability in response of virgin females suggests that mating enhances the female's initial attraction to a host.

## Neurotranscriptomics reveals several candidate genes that promote blood feeding

While sensitivity to host kairomones may modulate blood feeding, it is also possible that central brain circuitry contributes to it. We reasoned that cues from mating or signals from other tissues such as the midgut could act on brain circuitry to modulate blood feeding. To determine the molecular underpinnings of this behavioural modulation, we performed RNA sequencing from the central brains (brains without optic lobes) of mosquitoes at different stages of their reproductive cycle (consequently different states of blood-deprivation or -satiety). This included females at emergence (D0), 5 days after emergence (D5; sugar-fed, blood-deprived), virgins and mated females 1 day after a blood meal (D1$^{PBM}$, D1$^{PBM}$(m)), and females 1 day after oviposition (D1$^{PO}$(m)). Age-matched males were included for the earliest two time points (*Figure 3A*).

For each of our samples, 81–87% of about 20–30 million reads aligned to the *An. stephensi* genome (*Chakraborty et al., 2021*). The variance between our replicates was low as represented in the Principal Component Analysis (PCA) (*Figure 3B*). In PCA, older female brains, regardless of mating-, blood feeding-, or oviposition- status clustered together while the youngest female and male brains stood apart on PC1 and PC2 suggesting that age and sex contribute most significantly to the variance in our data (*Figure 3B*). The expression pattern of some hallmark genes showed the expected trends. Genes involved in eclosion behaviours – *eclosion hormone* (*Krüger et al., 2015*) and *partner of bursicon* (*Luo et al., 2005*) – showed the highest expression in the D0 samples and were not expressed later. Male-specific genes, *HMG-B-13-like*, were expressed only in males (*Hall et al., 2013*), and genes involved in vitellogenic stages of egg-development – *vitellogenins*, *ecdysone receptor*, and *E-20-monooxygenase* (*Attardo et al., 2005*) – were expressed only in post-blood meal samples (*Figure 3C*).

We next explored this data for insights into the molecular underpinnings of blood-feeding modulation. Reports suggest that anticipatory metabolic changes in female germline cells and a 'vitellogenin wave' in fat bodies can drive specific appetite in *Drosophila* (*Carvalho-Santos et al., 2020*) and *Ae. albopictus* (*Dittmer et al., 2019*), respectively. However, we did not find any evidence in the neurotranscriptomes for a similar anticipatory metabolic change that might drive blood feeding in the mosquito brain (*Figure 3—figure supplement 1B*). Instead, our data suggests altered carbohydrate metabolism after a blood meal, with the female brain potentially entering a state of metabolic 'sugar rest' while actively processing proteins (*Figure 3—figure supplement 1B*, *Figure 3—figure supplement 2*). However, physiological measurements of carbohydrate and protein metabolism will be required to confirm whether glucose is indeed neither spent nor stored during this period. We similarly plotted genes involved in lipid metabolism, neurotransmitters, neuropeptides and their receptors and found no significant patterns (*Figure 3—figure supplement 3*, *Figure 3—figure supplement 4*, *Figure 3—figure supplement 5*). In summary, contrary to our expectations, we saw no evidence in the neurotranscriptomes for anticipatory metabolic changes in the brain that might in turn drive blood-feeding behaviour. It is possible, however, that other forms of regulation that do not involve transcriptional changes could still influence metabolism.

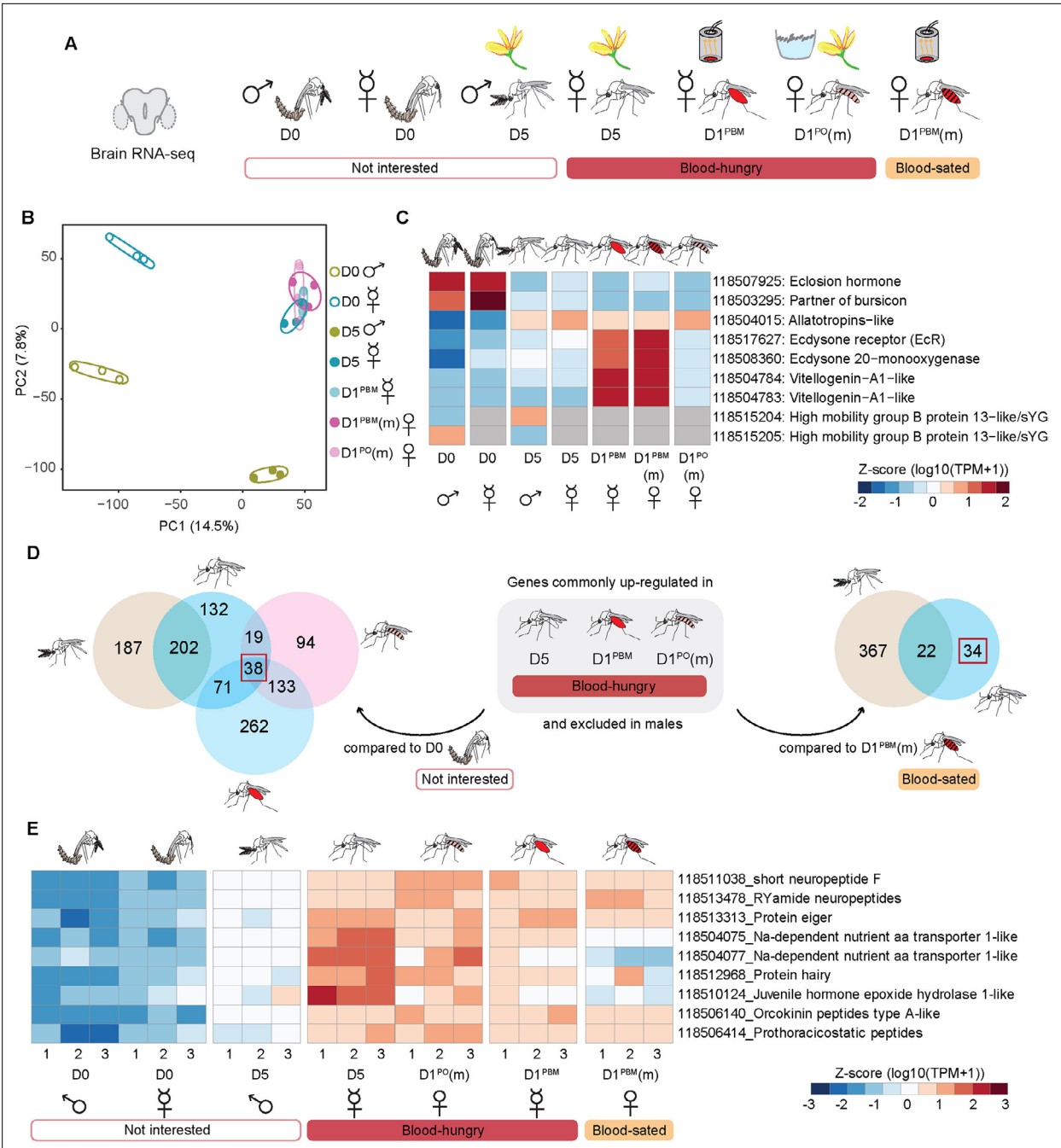

**Figure 3.** Neurotranscriptome of *An. stephensi*. (**A**) Central brains sampled from males and females at different ages and feeding preferences for bulk RNA-seq. Uninterested in blood: D0 males, D0 females, D5 males. Blood-hungry: D5, sugar-fed virgin females, D1PBM virgin females, and D1PO(m) mated females. Blood-sated: D1PBM(m) mated females. n=100 central brains per replicate; three biological replicates per sample, seven samples. (m): mated. (**B**) Principal Component Analysis (PCA) of the normalised counts-per-million. (**C**) Genes (listed on the right) with known expression patterns plotted across different samples. Average of the triplicate data are represented as z-score normalised log10 (TPM +1) values. Grey cells represent no data. (**D**) Identification of candidates potentially involved in promoting blood-feeding behaviour. Two types of comparisons were made: (left) genes commonly upregulated in the three blood-hungry conditions (see **A**) when compared against D0 females and excluded in males, and (right) genes upregulated in D5 sugar-fed virgin females when compared against D1PBM mated females (blood-sated) and excluded in males. Number of candidate genes identified from two different sets of analyses are boxed in red. (**E**) Nine candidate genes (listed on the right), shortlisted for further validation. RNA-seq data from central brain samples in triplicates are represented as z-score normalised log10 (TPM +1) values. (D0: day of emergence; D5: 5 days post-emergence, virgin; D1PBM: 1 day post-blood meal, virgin; D1PBM(m): 1 day post-blood meal, mated; D1PO(m): 1 day post-oviposition, mated).

The online version of this article includes the following figure supplement(s) for figure 3:

*Figure 3 continued on next page*

*Figure 3 continued*

**Figure supplement 1.** Differential expression of genes across various states of blood-hunger and -satiety.

**Figure supplement 2.** Differential expression of genes involved in carbohydrate metabolism across different tissues and states of blood-hunger and -satiety in *An. stephensi*.

**Figure supplement 3.** Differential expression of genes involved in lipid metabolism across different tissues and states of blood-hunger and -satiety in *An. stephensi*.

**Figure supplement 4.** Expression patterns of neuropeptides and neuropeptide receptors across different tissues and states of blood-hunger and -satiety in *An. stephensi*.

**Figure supplement 5.** Expression patterns of neurotransmitter-related genes in the central brain of *An. stephensi* across different states of blood-hunger and -satiety.

**Figure supplement 6.** Candidates potentially involved in promoting blood-feeding behaviour in *An. stephensi*.

In an alternate approach to identifying genes that might modulate blood feeding, we took the intersection of genes that were enriched in the three blood-hungry conditions (and absent in males). These comparisons were made against the two conditions that were not motivated to blood-feed: females at emergence and mated blood-fed females (*Figure 3D*). These shortlisted 38 and 34 genes respectively, gave us 68 unique candidates (excluding common genes and isoforms) that might promote blood feeding in *An. stephensi* (*Figure 3—figure supplement 6*). Notably, we did not find any genes to be differentially expressed between the two blood-fed conditions (virgins and mated), despite their contrasting behaviours (*Figure 3—figure supplement 1A*). The strongest signatures observed in these females were those related to metabolism (*Figure 3—figure supplement 1B*). It is possible that the differences lie in small populations of cells, which get obscured in the averaged transcriptomes of the entire brain.

## Neuropeptides *sNPF* and *RYamide* promote blood feeding in *An. stephensi*

We employed multiple, independent criteria while further shortlisting candidates from the above analysis: (1) genes that showed the expected expression pattern: low in newly emerged males and females; high in blood-hungry females; low in blood-sated females, (2) all neuropeptides, since these are known to regulate feeding behaviours, and (3) all genes related to juvenile hormone (JH) and 20-hydroxyecdysone (20E) signalling pathway, since they regulate development and reproduction in mosquitoes. This gave us nine candidates (*Figure 3E*): Two sodium-dependent amino acid transporters *118504075* and *118504077*. Transporters can sense nutrients and regulate feeding in *Drosophila* (*Park et al., 2016*; *Hundal and Taylor, 2009*). The neuropeptides included *Orcokinin (Ork)*, *Prothoracicostatic peptides (Prp)*, *RYamide (RYa)*, and *short neuropeptide F (sNPF)*, and a metabolic hormone *Eiger*, which responds to low-nutrient conditions in *Drosophila* (*Agrawal et al., 2016*). *Protein Hairy* and *Juvenile hormone epoxide hydrolase 1-like (JHEH)* are involved in the JH pathway (*Saha et al., 2016*; *Share and Roe, 1988*; *Figure 3E*).

Using dsRNA, we knocked each of these genes down and tested their blood-feeding behaviour. In each case, we ensured efficiency of gene knockdown post-hoc (*Figure 4A*). We first confirmed that the injection itself did not cause any behavioural changes (*Figure 4B*) and in all following experiments either *dsGFP*-injected or uninjected females were included as controls.

We saw no change in behaviour upon loss of *Eiger*, *JHEH*, and *Ork* function (*Figure 4—figure supplement 1A–C, A'-C'*). We were able to downregulate *Hairy* and *Prp* transcript levels by only 25–38%, hence cannot comment on their role in modulating blood feeding (*Figure 4—figure supplement 1D, E, D', E'*). Downregulation of the two transporters was efficient and resulted in a marginal reduction in blood feeding (*Figure 4—figure supplement 1F, G, F' and G'*). However, their combined knockdowns, though effective, did not show any significant change in behaviour (*Figure 4—figure supplement 1H, H'*).

Two neuropeptides – *sNPF* and *RYa* – showed about 25% and 40% reduced mRNA levels in the heads but the proportion of females that took blood meals remained unchanged (*Figure 4—figure supplement 2C, D, G and H*). We noticed, however, that many of these animals did not engorge and instead took smaller blood meals (*Figure 4—figure supplement 2F and J*). Given this, and since neuropeptides are known to act in concert (*Nässel and Zandawala, 2020*), we tested the possibility

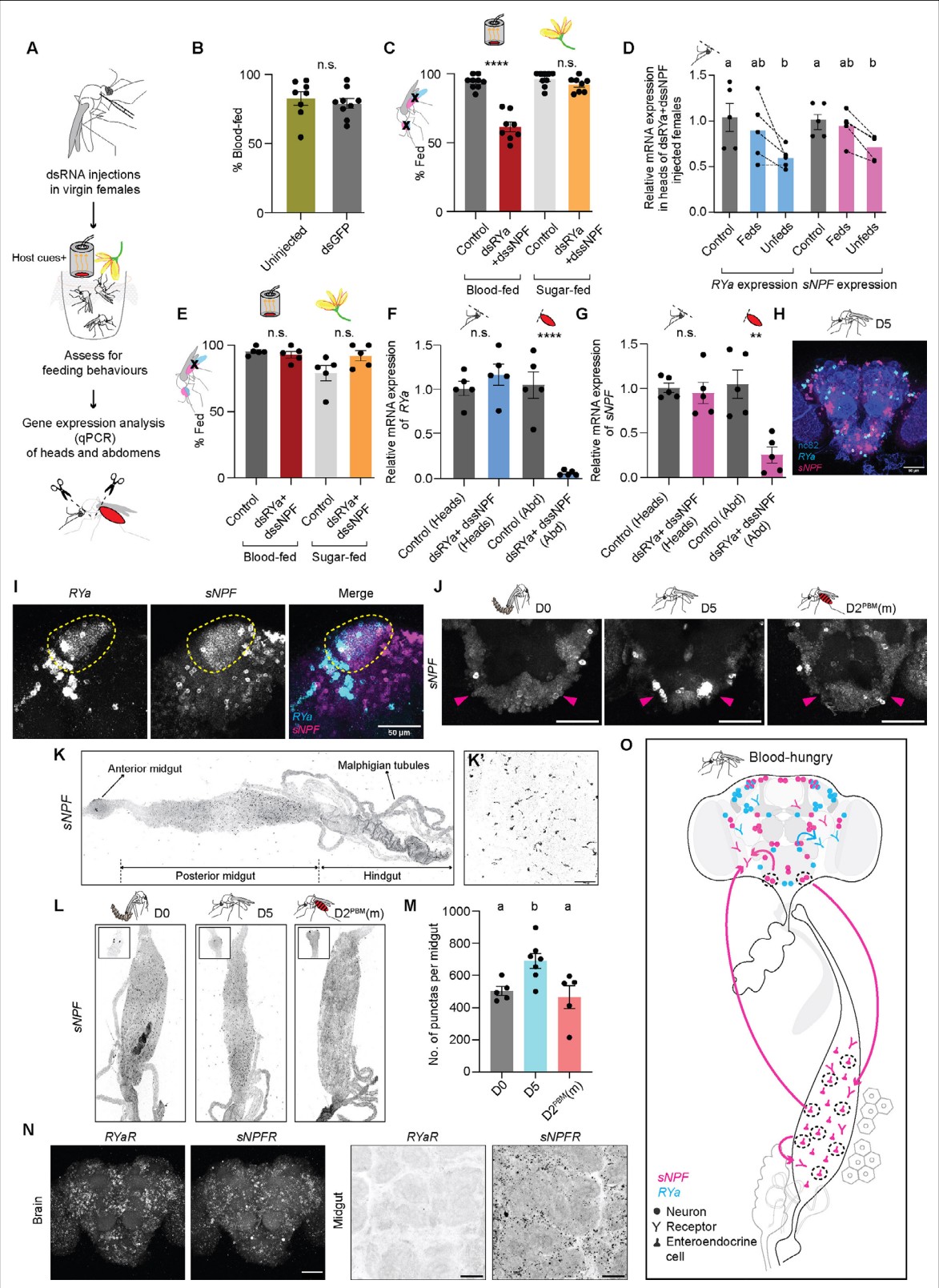

**Figure 4.** *short neuropeptide F* (sNPF) and *RYamide* (RYa) together promote blood feeding in *An. stephensi*. (**A**) Pipeline used for functional validation: dsRNAs were injected in adult virgins and tested for feeding behaviours. Knockdown efficiency was determined in heads and abdomens of fed and/ or unfed females after the behaviour. (**B**) Blood feeding in uninjected and *dsGFP*-injected females. 19–25 females/replicate, n=8–9 replicates/group. Mean ± SEM are plotted. Unpaired t-test; n.s.: not significant, p>0.05. (**C**) Blood- and sugar-feeding behaviour of females where both *RYa* and *sNPF*

*Figure 4 continued on next page*

*Figure 4 continued*

were knocked down in both the heads and abdomens. Controls: Uninjected females. 14–25 females/replicate, n=8–9 replicates/group. Mean ± SEM are plotted. Unpaired t-test; n.s.: not significant, p>0.05; ****p<0.0001. (**D**) Relative mRNA expressions of *RYa* and *sNPF* in the heads of *dsRYa +dssNPF-* injected blood-fed and unfed females assayed in (**C**). mRNA levels were compared to those in the uninjected controls. n=5 replicates/group. Mean ± SEM are plotted. One-way ANOVA with Holm-Šídák multiple comparisons test for relative *RYa* expression and Kruskal-Wallis with Dunn's multiple comparisons test for relative *sNPF* expression, p<0.05. Data labelled with different letters are significantly different. (**E**) Blood- and sugar-feeding behaviour of females when both *RYa* and *sNPF* were knocked down only in the abdomens. Controls: *dsGFP*-injected or uninjected females. 17–22 females/replicate, n=5 replicates/group. Mean ± SEM are plotted. Unpaired t-test; n.s.: not significant, p>0.05. (**F, G**) Relative mRNA expressions of *RYa* (**F**) and *sNPF* (**G**) in the heads and abdomens of *dsRYa +dssNPF-* injected fed females assayed in (**E**). mRNA levels in both tissues were compared to their respective *dsGFP*-injected or uninjected females controls. n=5 replicates/group. Mean ± SEM are plotted. Unpaired t-test; n.s.: not significant, p>0.05; **p<0.01; ****p<0.0001. (**H**) *RYa* (cyan) and *sNPF* (magenta) HCR *in situ* hybridisation in central brain of 5-day-old sugar-fed virgin female. nc82 (blue). Scale bar, 50 µm. (**I**) Co-expression of *RYa* (cyan) and *sNPF* (magenta) in mushroom body Kenyon cells of 5-day-old sugar-fed virgin female. Scale bar, 50 µm. (**J**) *sNPF* expression in the SEZ of females uninterested in blood (D0), blood-hungry females (D5) and blood-sated females (D2$^{PBM}$(m)). Magenta arrow marks the novel *sNPF* cluster only in the blood-hungry (D5) condition. Scale bar, 50 µm. (**K**) *sNPF* HCR *in situ* hybridisation in gut of 5-day-old sugar-fed virgin female. Higher magnification image is shown in K'. Scale bar, 50 µm. (**L**) *sNPF* expression in the posterior midguts of females uninterested in blood (D0), blood-hungry females (D5) and blood-sated females (D2$^{PBM}$(m)). Note expression in enteroendocrine cells (**K'**) and in two cells in the anterior midgut (**L**, insets). (**M**) No. of *sNPF*-positive cells. n=5–7 guts per condition. Mean ± SEM are plotted. One-way ANOVA with Holm-Šídák multiple comparisons test, p<0.05. Data labelled with different letters are significantly different. (**N**) *sNPF receptor* (*sNPFR*) and *RYa receptor* (*RYaR*) HCR *in situ* hybridisation of central brain (left) and midgut (right) of 5-day-old sugar-fed virgin female. Scale bar, 50 µm. (**O**) Proposed model: *RYa* and *RYaR* (cyan) are expressed only in the brain, while *sNPF* and *sNPFR* (magenta) are expressed both in the brain and the gut. Increased *sNPF* levels in both the tissues (dashed circles) promote a state of blood-hunger, which may drive feeding behaviour either by *sNPF's* action in the two tissues independently or via a communication between them. This happens in the context of *RYa* signalling in the brain. (D0: day of emergence; D5: 5 days post-emergence, virgin; D2$^{PBM}$(m): 2 days post-blood meal, mated).

The online version of this article includes the following video and figure supplement(s) for figure 4:

**Figure supplement 1.** Screening of the shortlisted candidates, potentially involved in promoting blood-feeding behaviour in *An. stephensi* via dsRNA-mediated gene knockdown.

**Figure supplement 2.** Abdominal knockdown of *RYa* or *sNPF* alone does not contribute to the blood-feeding behaviour in *An. stephensi*.

**Figure supplement 3.** Transcripts of neuropeptides *sNPF*, *RYa* and their receptors are expressed in *An. stephensi* central brain.

**Figure supplement 4.** Transcripts of *sNPF* and *sNPFR* are expressed in *An. stephensi* posterior midgut.

**Figure 4—video 1.** *sNPF* expression in the central brain of females uninterested in blood (D0), blood-hungry females (D5), and blood-sated females (D2$^{PBM}$(m)), related to **Figure 4J**.

https://elifesciences.org/articles/108625/figures#fig4video1

that *sNPF* and *RYa* might act together to regulate blood-feeding behaviour. Indeed, their simultaneous dsRNA injections resulted in about 40% of the animals not taking blood meals (**Figure 4C**). Since RNA interference (RNAi) can be variable, and each female is an independent experiment, we pooled the females that took blood meals and those that didn't separately before quantifying mRNA from them. We noted knockdowns in the heads of females that did not take blood meals suggesting that downregulating these genes caused the suppression of blood feeding (**Figure 4D**). Importantly, since these females were offered both blood and sugar, we noted that their sugar feeding was unaffected (**Figure 4C**).

Since RNAi results in systemic knockdowns and neuropeptides are known to influence behaviour from their action in non-neuronal tissues such as the gut (**Malita et al., 2022**; **Dou et al., 2024**; **Nässel and Zandawala, 2020**), we modified our dsRNA injection protocol to knock *sNPF* and *RYa* down in the abdomen while leaving their levels in the head unaltered (**Figure 4F, G**, **Figure 4—figure supplement 2B, M, N, P and Q**) (see Materials and methods for details). Abdomen-specific knockdown had no effect on blood feeding (**Figure 4E**, **Figure 4—figure supplement 2L, O**), indicating that abdominal *sNPF* and *RYa* alone cannot influence blood feeding and their action in the brain is necessary. In contrast, since every head knockdown of these neuropeptides also abolished their abdominal expression (**Figure 4—figure supplement 2D, E, H and I**), we cannot rule out the possibility that both brain- and gut-derived neuropeptides modulate blood feeding. Indeed, since such neuropeptide-mediated gut-brain control of feeding is commonly seen (**Nässel and Zandawala, 2020**), we propose a model, consistent with our data, where brain-derived *sNPF/RYamide* is essential to trigger blood feeding, while gut-derived peptide acts as a modulatory feedback signal. This remains to be tested with cell-specific tools that are currently not available for this species.

To determine in which cells *sNPF* and *RYa* act, we used hybridisation chain reaction (HCR; *Choi et al., 2014*; *Choi et al., 2018*) *in situ*. We found that their transcripts are expressed in several non-overlapping neuronal clusters distributed across many regions in the brain (*Figure 4H*, *Figure 4—figure supplement 3A*) and only in the Kenyon cells of the mushroom body were they co-expressed (*Figure 4I*, *Figure 4—figure supplement 3B*). This is reminiscent of their expression in *Drosophila* and *Bombyx* (*Nässel et al., 2008*; *Roller et al., 2016*; *Veenstra and Khammassi, 2017*). Importantly, we found a cluster of cells in the SEZ that expresses the *sNPF* transcript only in the blood-hungry state (*Figure 4J*, *Figure 4—video 1*). Its transcripts were either absent or diminished from this cluster in newly emerged or sated females. Since the RNAi experiments suggested the abdominal involvement of these neuropeptides, we also looked for their expression in the midgut where we could detect only *sNPF* transcripts, and not those of *RYa* (*Figure 4K, K'*, *Figure 4—figure supplement 4A, B and E*). Interestingly, *sNPF* transcript expression was also modulated in the midgut enteroendocrine cells (EECs) – hormone-producing cells of the gut. In these cells, *sNPF* transcripts were specifically increased in blood-hungry females when compared to newly emerged or sated females (*Figure 4L and M*).

Neuropeptides work by acting on their receptors to trigger downstream signalling cascades in the cells that express the receptors. We identified putative receptors for *sNPF* and *RYa* (see Materials and methods) and found that both were expressed in several overlapping cell clusters in the brain (*Figure 4N*, *Figure 4—figure supplement 3C and D*). In the midgut, however, we could only detect *sNPFR* and not *RYaR* (*Figure 4N*, *Figure 4—figure supplement 4C, D and F*). Since these receptors were identified based on homology, their functional validation will be necessary to implicate them in blood-feeding behaviour.

Together, these data are consistent with a model where increased *sNPF* expression in both the brain and midgut might drive the blood-feeding behaviour, but that this occurs in the context of *RYa's* action in the brain (*Figure 4O*).

## Discussion

Here, we report the blood- and sugar-feeding behaviours of *An. stephensi*. While sharing key features with other mosquito species, *An. stephensi* exhibits distinct regulatory patterns. Unlike *Ae. aegypti*, it does not require mating to initiate blood feeding, although the presence of males is necessary to sustain the enhanced state. In both species, blood feeding is suppressed after a blood meal and until oviposition, but this suppression is mating-dependent in *An. stephensi*. Since unmated *Ae. aegypti* females did not blood-feed in our assays, we cannot comment on whether this suppression also requires mating.

Mosquito feeding behaviour is highly context-dependent, shaped by physiological state, environment, and experimental design (*Scott and Takken, 2012*). Our comparisons between *Ae. aegypti* and *An. stephensi* were conducted under standardised conditions. While direct comparisons across studies must be made cautiously, patterns observed under particular conditions can offer meaningful insights. For example, virgins of other *Anopheles* species readily blood-feed (*Charlwood et al., 2003*; *Ye et al., 2022*; *Stone et al., 2011*; *Scott and Takken, 2012*), suggesting that this is typical of the genus, whereas *Ae. aegypti* virgins do so only under specific conditions and these responses vary among strains (*League et al., 2021*; and references within). Whether blood-fed *Aedes* virgins suppress subsequent feeding is likely also context-dependent with reports suggesting both that they (*Lavoipierre, 1958*; *Judson, 1967*) and do not (*Klowden and Lea, 1979b*). Post-blood meal suppression of host seeking – well characterised and robust in mated *Aedes* – appears weaker in mated *Anopheles* and can be lifted prior to oviposition (*Takken et al., 2001*). Indeed, we too observed a stronger post-blood meal suppression of feeding in *Ae. aegypti* than in *An. stephensi* (compare *Figure 1C* to *Figure 1D*). Few studies have investigated this suppression in anopheline virgins, but when they do, they report a weak suppression (*Rowland, 1989*). Taken together, *An. stephensi* displays characteristics of both culicine and anopheline feeding patterns – its virgin feeding patterns align more closely with other anophelines, yet, like mated *Ae. aegypti*, it still exhibits clear suppression of blood feeding until oviposition.

What mechanisms might underlie such mating-induced modulation of blood feeding? In *Drosophila melanogaster*, mating induces a suite of behavioural changes – including increased protein, salt and sugar intake – that are mediated by male-derived sex peptide that acts on central brain circuits in

the female brain (*Carvalho et al., 2006*; *Vargas et al., 2010*; *Ribeiro and Dickson, 2010*; *Walker et al., 2015*; *Laturney et al., 2023*). These changes are anticipatory in nature rather than a homeostatic control of feeding behaviour to meet reproductive demands (*Walker et al., 2015*; *Laturney et al., 2023*). In mosquitoes, the molecular mechanisms driving post-mating behavioural changes are not likely to be the same (*Amaro et al., 2024*). Instead, mating-derived modulators such as HP1 in *Ae. aegypti* (*Duvall et al., 2017*; *Naccarati et al., 2012*) and male-deposited steroid hormone (20-hydroxyecdysone) in *An. gambiae* (*Peng et al., 2022*; *Gabrieli et al., 2014*) have been shown to induce sexual refractoriness. Whether they also shape feeding decisions remains unknown.

We have identified two neuropeptides – *RYa* and *sNPF* – that act synergistically to promote blood feeding in *An. stephensi*. Our data are consistent with a model where an increase in *sNPF* levels in the brain – likely released by a cluster of neurons in the SEZ – as well as in the midgut, promotes a state of blood-hunger. Whether this reflects gut-to-brain communication or parallel tissue-local computations to modulate behaviours remains unclear, especially since we detected the transcript of both *sNPF* and its receptor in both tissues. Inter-organ signalling is well-established in feeding regulation (*Nässel and Zandawala, 2020*; *Zhao et al., 2025*). For example, gut-derived NPF in flies can influence brain circuits to increase protein intake (*Malita et al., 2022*). More broadly, feeding behaviour is likely regulated by a distributed network of hormonal and neuronal cues across multiple organs. Signals from the ovary, fat body, and other nutrient-sensing tissues may also contribute, acting through both central and peripheral circuits. Many of these inputs remain to be identified.

It is possible that *sNPF* promotes blood feeding independent of *RYa*. However, we were unable to knock down *sNPF* completely to test this. Its synergistic requirement with *RYa*, despite no apparent modulation of *RYa* or its receptor, could result from a general state of hunger that *RYa* promotes, in the context of which, we were able to uncover the blood-feeding role of the diminished *sNPF*. Since many neurons co-express these receptors, it is possible that neurons downstream of *sNPF* or *RYa* expressing neurons integrate the two inputs to modulate behaviour.

We note that our findings differ from those reported in *Ae. aegypti*. In *Aedes*, *sNPF* and *RYamide* act as satiety brakes—peptide or receptor activation curbs host seeking (*Dou et al., 2024*; *Liesch et al., 2013*; *Christ et al., 2017*) —whereas in *Anopheles*, our knockdowns show that the same neuropeptides act as a hunger signal, stimulating blood feeding. While these differences are interesting, in the context of their evolutionary histories, and the known differences in how neuropeptides can act differently across species, they are not entirely unexpected. *Aedes* and *Anopheles* lineages diverged ≈150–200 million years ago (*Krzywinski et al., 2006*), and the two genera have since diverged in almost every facet of biology - diel activity, host preference, feeding and oviposition sites, egg desiccation tolerance, genome size, and chemosensory repertoires among other traits. Neuropeptide pathways are known to drive opposing behaviours in different species: *sNPF* is known to be orexic in *D. melanogaster*, *A. mellifera*, *L. decemlineata*, *B. mori*, and *P. americana*, while it has anorexic effects in *S. gregaria* (*Fadda et al., 2019*). Against this backdrop, it is unsurprising that the same neuropeptide pathway can drive opposite behaviours in *Aedes* and *Anopheles*. These comparative studies underscore a key lesson for vector control strategies in general. A strategy that is effective in one mosquito species may inadvertently boost vectorial capacity in another. Behavioural interventions, chemical modulators, and even gene drives therefore require rigorous cross-species validation because results from one mosquito species cannot be assumed to hold true for another.

## Materials and methods
### Mosquito culture
*Anopheles stephensi* (Indian strain; Bangalore strain, TIGS-1) and *Aedes aegypti* (Bangalore collection) were reared in BugDorm cages and maintained at 26–28°C, 70–80% relative humidity with a photo-period of 12 hr light: 12 hr dark. All the behavioural assays were performed at these conditions of temperature and humidity. Larvae were fed on a mixture of yeast and dog biscuit (1:3 ratio) and cultured in trays in appropriate densities to prevent overcrowding. Adult mosquitoes were provided *ad libitum* access to 10% sugar solution (8% sucrose +2% glucose +0.05% methyl paraben +5% vitamin syrup [Polybion SF Complete, Merck Ltd.]) at all times, unless specified otherwise. Females were provided O+ve human blood, secured periodically from blood banks, both for general maintenance and feeding assays.

### Regulatory permissions

This project was approved by the Institutional Human Ethics Committee (Ref: inStem/IEC-22/02) and the Institutional Biosafety Committee (TIGS/M-7/8/2020–1).

### Blood-feeding assay: first blood meal, 0–120 hr post-emergence (D0–D5)

All *An. stephensi* feeding experiments were performed from ZT12-ZT15 and *Ae. aegypti* experiments from ZT10 to ZT12 (Zeitgeber time 0 is defined as the time of lights ON).

0–7 hr-old mosquitoes were cold anaesthetised and collected during the day in 1100 ml paper cups covered with a net and secured with a rubber band. These were maintained at the conditions of temperature and humidity described above. To collect virgins, males were separated at the time of collection. 20 females with males (1:1 ratio) or without males were collected for each trial and were aged appropriately with *ad libitum* access to 10% sugar solution (*Figure 1—figure supplement 1A*).

Females were assayed for their blood-feeding behaviour from the day of emergence (D0, 0–7 hr-old) to 5 days of age (D5), every 24 hr. They were provided constant access to sugar, except when they were offered blood. A detailed schematic has been shown in *Figure 1—figure supplement 1A*. On the day of the test, appropriately aged females were offered a blood meal via a Hemotek feeder (Hemotek Ltd), which was maintained at 37 °C and perfumed with human skin odours. It has been reported that female mosquitoes need to be 'activated' to take blood meals. This involves the presentation of $CO_2$ (and other host cues) to the females, likely because $CO_2$ has been shown to gate temperature and visual cues (*McMeniman et al., 2014*; *Wolff and Riffell, 2018*). So, prior to offering the blood meal, the mosquitoes were 'activated' by the presentation of a hand placed above the cup for 5 min, followed by three exhalations from the experimenter. Additionally, blood was supplemented with 1 mM ATP (Sigma A6419) before loading the feeder (*Figure 1B*). Females were allowed to feed for 60–90 min in dark and abdomens were scored visually for presence of blood, irrespective of the amount ingested (inset, *Figure 1—figure supplement 1A*). For D0, no sugar was provided at the time of collection. Females were allowed to acclimatise for at least 30 min in the cups and then offered a choice of both sugar and blood simultaneously, to assess their preference for blood soon after emergence (*Figure 1—figure supplement 1A*).

In the case of co-housed females, males and females were housed together from the time of collection until the day of the experiment. Males were separated post-hoc and spermathecae of the females were dissected to score for the presence or absence of sperm (*Figure 1G*). All the assays with virgin and co-housed females were performed in parallel by a single experimenter. Mosquitoes from at least four independent batches were tested for each day of the experimental timeline.

For *Ae. aegypti*, both virgin and mated females (mating status was confirmed via post-hoc dissection of spermathecae and all co-housed females were found to be mated) were tested for first blood meal at only 8 days post-emergence (D8, *Figure 1C*). Behavioural experiments were performed the same way as described for *An. stephensi*.

### Blood-feeding assay: second blood meal, 24–96 hr post-first blood meal (D1$^{PBM}$-D4$^{PBM}$)

To ensure that the second blood meal is not driven due to an incomplete first blood meal, we selected only fully engorged females to test for subsequent feedings (*Figure 1—figure supplement 1A*). 5-day-old *An. stephensi* (both virgins and co-housed with males) and 7–9 days-old *Ae. aegypti* (mated, as previously determined) were fed on blood and fully fed females were separated and collected in paper cups as described above. As a female takes about 2 days to digest blood, the second blood meal ingested 24 hr post-first blood meal (D1$^{PBM}$) was detected by the presence of Rhodamine B (Sigma R6626), which fluoresces in the red range (*Figure 1—figure supplement 1A*). Spiking the second blood meal with Rhodamine B (0.04 μg/μL of blood) allowed visualisation of the freshly ingested blood without affecting the feeding efficiency (*Figure 1—figure supplement 1B–D*). To assess feeding at subsequent days (D2$^{PBM}$-D4$^{PBM}$), abdomens were dissected to differentiate between the bright red freshly ingested blood from the dark and clotted first blood meal. In the case of co-housed *An. stephensi* females, spermathecae were not dissected to confirm mating as females were presumed to be mated when collected for this assay (5 days-old and co-housed with males for the entire duration from the time of emergence), as previously determined in *Figure 1G*.

To confirm whether mating suppresses subsequent feeding (*Figure 1H*), fully fed virgins were collected and allowed to mate for 3–4 days with aged-matched males. No males were introduced in the virgin-only controls. Blood-feeding assay was performed as described above. Spermathecae were dissected post-hoc to determine the mating status and unmated females were not considered for the final analysis. Datasets with less than 80% total mating were discarded.

## Blood-feeding assay: post-oviposition, 24–48 hr post-egg laying (D1$^{PO}$-D2$^{PO}$)

For blood-feeding behaviour post-oviposition, fully fed females were separated and a water cup lined with moist paper (ovipositor) was provided for egg laying 2 days post-blood meal (determined in a separate assay described below, *Figure 1—figure supplement 1A, E*). Females were allowed to lay eggs overnight. The following morning, non-gravid females were visually identified by their lean bellies and separated in cups. Blood-feeding assay was performed the same evening for 1 day post-oviposition (D1$^{PO}$) and the following day (D2$^{PO}$), as described above. Females were dissected post-hoc and scored for absence of mature ovaries, as a sign of successful oviposition. Gravid females were not considered for the analysis.

## Determination of oviposition timing for *An. stephensi*

To determine the day of maximum oviposition, 5- to 6-day-old females co-housed with males were fed on blood, as described above. Fully fed females were separated and collected in BugDorm cages in groups of 10 females/cage. An ovipositor was provided the same day and females were allowed to lay eggs overnight. Eggs laid were counted the next day and a fresh ovipositor was provided. This cycle was repeated for the next 5 days. Total eggs laid each day were divided by the number of females present at the time of counting (*Figure 1—figure supplement 1E*).

## Sugar-feeding assay for *An. stephensi*

As mosquitoes feed on sugar in small bouts, as opposed to a replete blood meal, we assessed sugar-feeding behaviour of females in windows of 24 hr for each test day. 0- to 7-hr-old mosquitoes were collected during the day, separated in cups and aged appropriately. They were given access to 10% sugar solution at all times, as described above, except for the day of emergence (D0) when no sugar was provided at the time of collection (similar to the blood-feeding assay performed at D0).

To assess their sugar-feeding behaviour on the day of emergence (D0), 0- to 7-hr-old females were acclimatised in the cups for 1 hr and sugar spiked with Rhodamine B (1 mg/25 ml of sugar; Sigma R6626) was presented for 2.5–3 hr. The fluorescence of Rhodamine B in the red channel allowed visualisation of even the tiniest amounts of sugar ingested by a female (*Figure 1—figure supplement 1G*). For D1 feeding, 3- to 10-hr-old females (with access to normal sugar so far) were then provided with Rhodamine B-sugar and allowed to feed on it for 24 hr. For D2 feeding, 27- to 34-hr-old females (with access to normal sugar so far) were fed on Rhodamine B-sugar for the next 24 hr and scored as D2. This cycle was repeated until D5. In the case of co-housed females, males were separated post-hoc and spermathecae of the females were dissected to score for the presence or absence of sperm (*Figure 1—figure supplement 1H*).

To assess sugar feeding post-blood meal, females fully fed on blood were collected and separated in cups, as described above. They were allowed to feed on Rhodamine B-sugar for 24 hr following a blood meal and scored as D1$^{PBM}$. The next set of females (with access to normal sugar so far) were then provided with Rhodamine B-sugar from 24 hr to 48 hr post-blood meal and scored as D2$^{PBM}$. This cycle was repeated until 4 days post-blood meal (D4$^{PBM}$, *Figure 1E*).

For sugar feeding post-oviposition, blood-fed females were provided an ovipositor 3 days later and were forced to oviposit during daytime by artificially creating dark conditions. Oviposited females (non-gravid) were then visually identified, separated and presented with Rhodamine B-sugar for the following 24 hr and scored as D1$^{PO}$. The next set (with access to normal sugar so far) was then provided with Rhodamine B-sugar, 24–48 hr post-oviposition and scored as D2$^{PO}$ (*Figure 1E*). For each day tested, females were dissected post-hoc and scored for absence of mature ovaries, as a sign of successful oviposition. Gravid females were not considered for the analysis.

## Dual choice assay of blood and sugar

Mosquitoes were collected as described above. Males and females were co-housed for 4–6 days with *ad libitum* access to sugar. To assess the choice between sugar and blood, mosquitoes were either

given continuous access to normal sugar (sugar-sated) or starved with water 24 hr prior to the assay (sugar-starved). At the time of the assay, post-activation (as described above), blood, and Rhodamine B-spiked sugar (1 mg/25 ml of sugar, Sigma R6626) were provided in parallel. After 60 min, abdomens were scored visually for ingestion of blood under white light, and under fluorescent microscope (red channel) for ingestion of sugar (*Figure 1—figure supplement 1F*). Co-housed males served as an internal control for sugar feeding. As males are unable to feed on blood, they were only scored for presence or absence of Rhodamine B-spiked sugar. Results are presented in *Figure 1F*.

## Statistical analysis of feeding behaviours

Statistical analysis was performed using R (version 4.4.2) and GraphPad Prism (v10.2.0). Blood- and sugar-feeding behaviours of virgins and co-housed females in *Figure 1D and E* were analysed using Generalized Linear Mixed-effects Models (GLMMs) with a binomial distribution and logit link function. The models were fitted using the 'glmer' function from the 'lme4' package in R. The probabilities of blood- or sugar-feeding, both pre- and post-blood meal were modelled as a function of Day (age) and Group (virgin vs. co-housed with males), including their interaction term to test whether group differences varied across days. Replicate identities (BioID) and larval batches (only for first blood meal in sugar-fed females) were included as random factors to account for replicate-level and batch-level variations respectively. Post-hoc analyses were conducted using estimated marginal means (emmeans package) for age-effects within groups (p-value adjusted for multiple comparisons using Tukey's method) and pairwise comparison between groups at each day (p-value adjusted using Bonferroni method). To test whether mating success altered the sugar- and blood-feeding propensity of females co-housed with males, the probabilities of feeding were modelled only the co-housed data, as a function of Day (age) and Mating (yes or no), including their interaction term and accounting for variation due to replicate IDs (BioID). Statistical significance was set at α = 0.05 for all analyses. Complete model outputs are provided in *Supplementary file 1*.

Statistical analyses for data in *Figure 1C, F and H* were performed using GraphPad Prism (v10.2.0) with appropriate tests as indicated in figure legends.

## Y-maze olfactory assay for host-seeking behaviour in *An. stephensi*

A 30 inch Y-maze was constructed as per the WHO guidelines (https://www.who.int/publications/i/item/9789241505024). Each arm of the Y-maze had a trapping port, while the stem was attached to a holding port. Each port had a mesh screen at one end, while the other was fitted with a sliding door to trap or release mosquitoes. The holding port was attached to a fan which pulled air from the two arms, over the mosquitoes, and out at a velocity of 0.3–0.6 m/s. Optic flow information (alternate black and white stripes) was placed underneath the Y-maze to act as a visual guide for navigation.

Following conditions were tested: D0 virgins, D5 sugar-fed virgins, D5 sugar-fed mated females (D5(m)), 3 days post-blood meal virgins (D3$^{PBM}$), 3 days post-blood meal mated females (D3$^{PBM}$(m)), and 2 days post-oviposition mated females (D2$^{PO}$(m)). For mated conditions, females were co-housed with males for 5 days post-emergence and assumed to be mated based on our previous findings (*Figure 1G*). For each condition, mosquitoes were collected and provided *ad libitum* access to sugar as described above. Females were transferred to holding ports 7–8 hr prior to the assay and were starved during this time with only access to water. For each run, mosquitoes were acclimatised for 5 min: 2 min with clean air and 3 min with an experimenter presenting a hand at one of the arms and simultaneously exhaling at resting rates. To account for disruption in airflow, the hand of a mannequin was placed at the other arm (control). Post-acclimatisation, the door of the holding port was opened to release the mosquitoes and they were allowed to make a choice for 5 min in the presence of continued host cues. The ports were then closed and mosquitoes in each arm counted. All the mosquitoes trapped in a port, sitting on the door or resting on the half-length of the Y-arm were considered as 'attracted' to that stimuli (host or control, *Figure 2A*). Participation percentage was also calculated as number of mosquitoes that passed half-way through the Y-stem/total number of mosquitoes released (*Figure 1—figure supplement 1I*).

Several controls were included to avoid experimental artefacts. Each experimental day started with a blank trial (where no stimuli were presented on either arm of the Y-maze) and the side of the host cues presentation was alternated with each trial to eliminate side biases. Trials for blood-fed virgins and mated females were done both in parallel and on multiple days to eliminate the influence of

external factors which might be present on a given day. When testing blood-fed females, age-matched sugar-fed females (blood-hungry) were included wherever possible as positive controls. These females consistently showed attraction to host cues, as expected. Lastly, we made sure that at least 50% of the mosquitoes had made either choice in all runs, except for D0 (newly emerged females) and blank runs. In the rare cases that this was not true, we did not consider those for the analysis. Statistical analysis was performed using GraphPad Prism (v10.2.0). To compare the attraction of females to host cues and control in *Figure 2B*, each time-point was treated as an independent experiment and p-values were computed for each pair-wise comparison.

## Bulk RNA sequencing of central brains and data analysis

For bulk RNA sequencing, central brains (ie. brains without optic lobes) were dissected from the following conditions: D0 and D5 sugar-fed males; D0 and D5 sugar-fed virgin females; 1 day post-blood meal virgin (D1$^{PBM}$) and mated females (D1$^{PBM}$(m)) and 1 day post-oviposition mated females (D1$^{PO}$(m)). For post-blood meal conditions, 5-day-old females were fed on blood and fully fed females were separated for sample preparation. Females co-housed with males for 5 days were assumed to be mated based on our previous findings (*Figure 1G*). Three replicates from independent batches were prepared for each condition.

Mosquitoes were cold anaesthetised before dissections. Central brains (with optic lobes removed) were dissected in RNase-free 1 X PBS and immediately placed into ACME solution (13:3:2:2 of water, methanol, acetic acid, and glycerol) on ice, which mildly fixes the tissues (*García-Castro et al., 2021*). For each condition, 100 brains were dissected per replicate. The mildly fixed brains were washed with cold 1 X PBS and centrifuged at 13,000 rpm for 5 min at 4 °C to remove ACME. Brains were then homogenised in TRIzol (Invitrogen), flash-frozen, and stored at –80 °C until RNA extraction. RNA was extracted using Direct-zol RNA MicroPrep (Zymo Research) columns. RNA quantity and quality were assessed using the Thermo Fisher Qubit 4 and RNA 6000 Nano kit Bioanalyser, respectively.

The cDNA library was prepared for each sample using the NEBNext Ultra II Directional RNA Library Prep kit (NEB E7765L), with an average fragment size of 250 bp. Some of the library samples were subjected to size and quality checks using a bioanalyser. Paired-end sequencing was carried out on the Illumina Hiseq 2500 sequencing platform, generating 2×100 bp reads. The library preparation and sequencing were performed at the Next Generation Genomics Facility (NGGF) at Bangalore LifeScience Cluster (BLiSC), India.

Post-sequencing, samples were demultiplexed and delivered as raw Fastq files. After passing the quality check using the FastQC tool, the reads were mapped onto the reference *An. stephensi* transcriptome (NCBI RefSeq assembly GCF013141755.1; *Chakraborty et al., 2021*) using Kallisto v0.43.1 (*Bray et al., 2016*) with default parameters. 81–87% of reads aligned to the mosquito transcriptome.

Transcript-level abundance was quantified using Kallisto (*Bray et al., 2016*) and gene-level summarisation was performed using the Tximport function in R (*Soneson et al., 2016*), with transcript isoform information retained (*Supplementary file 2*). Using the edgeR package (*Robinson et al., 2010*), the raw counts were converted to counts per million (CPM) and log2 transformed before data filtering (genes with less than 1 CPM in at least three or more samples were filtered out) and normalisation, using the trimmed mean of M-values (TMM) method. Principal component analysis (PCA) was performed using the prcomp function in R and visualised using the ggplot function. Data was checked for batch effects using gPCA (*Reese et al., 2013*) in R and the test statistic 'delta' was found to be 6.957686e-05, with a p-value of <0.001, indicating no batch-effects.

### Differential gene expression analysis

Differential expression analysis was performed using the *limma* package (*Ritchie et al., 2015*). A design matrix was created to specify the sample conditions, and the Voom function was used to estimate the mean-variance relationship across genes. A linear model was then fitted to the data using the mean-variance relationship. Bayesian statistics were calculated for the data by creating a contrast matrix to extract the linear model fit. To identify differentially expressed genes (DEGs), the 'decideTests' function was used by applying a fold change threshold of 2 (log2fc = 1) and a FDR (false discovery rate) adjusted p-value cut-off of 0.05. The number of genes identified to be upregulated and downregulated in each pair-wise comparison are shown in *Figure 3—figure supplement 1A* and the gene lists are provided in *Supplementary file 3*.

To plot transcripts per million (TPM) values as heatmaps, the Fastq files for the central brain samples were mapped onto the reference *An. stephensi* transcriptome and abundance was quantified using Kallisto (*Bray et al., 2016*) the same way as described above. Gene-level summarisation was done using Tximport (*Soneson et al., 2016*) as described above, except all the isoforms of a given transcript were mapped to a single gene (*Supplementary file 2*). The resultant TPM values were first filtered for TPM >0.1 in each sample. Next, the genes that were not quantified in at least two of the three replicates of each sample group were removed. The filtered dataset was then log10 transformed and heatmaps were plotted using the pheatmap function in R. To construct heatmaps of genes involved in carbohydrate and lipid metabolism (*Figure 3—figure supplement 2*, *Figure 3—figure supplement 3*), key enzymes of the respective canonical pathways were manually curated based on the existing literature. Gene annotations for neuropeptides and neuropeptide receptors were curated from published sources (*Figure 3—figure supplement 4*; *Strand et al., 2016*; *Matthews et al., 2016*; *Riehle et al., 2002*). Fat body, midgut, and ovary data (SRX620223, SRX618937 and SRX620224; BioProject PRJNA253267) for 2- to 4-day-old, sugar-fed *An. stephensi* females were taken from *Prasad et al., 2017*. Fastq files were downloaded from the SRA explorer and processed similarly (*Supplementary file 2*).

## Gene set enrichment analysis

eggNOG mapper tool 2.1.9 (*Cantalapiedra et al., 2021*) (parameters: 70% identity and minimum 70% query coverage) was used to find functionally annotated orthologs of the identified DEGs in *An. gambiae*. *An. gambiae*-specific orthologs were fed into the GOSt tool from gProfiler2 (*Kolberg et al., 2020*) toolset for functional enrichment analysis. A custom gene list comprising all the genes identified in our RNA-seq experiment was provided as a background instead of the whole genome. Using this background, an ordered query was run with an FDR adjusted p-value threshold of 0.05 to determine which GO terms (Biological Processes) were over-represented. This was done for all sets of DEGs and the top 10 terms of each set were plotted using ggplot function in R (*Figure 3—figure supplement 1B*).

## **dsRNA-mediated gene knockdown**

To synthesise the double-stranded RNA (dsRNA), coding regions (500–600 bp) of candidate genes were amplified from cDNA of *An. stephensi* females using forward and reverse primers with T7 promoter sequence at 5' ends. The gel purified fragments were then used as a template for *in vitro* transcription reactions using Megascript RNAi kit (Invitrogen) or HighYield T7 RNAi Kit (Jena Biosciences) as per the manufacturers' protocols. The dsRNAs were purified using the phenol-chloroform method and pellet dissolved in nuclease-free water. The dsRNA for green fluorescent protein (GFP) was synthesised from plasmid pAC5.1B-EGFP (Addgene 21181) to be used as a control for injection-related behavioural changes.

Virgins were collected for injections as described above. The day and amount of dsRNA to be injected were determined for each target gene individually. *dsEiger*: 2.5–3.6 μg injected per female in 24-hr-old virgins; *dsHairy*: 2–4.3 μg injected per female in 24-hr-old virgins; *dsJHEH*: 2.5–3 μμg injected per female in 72-hr-old virgins; *dsPrp*: 1.5–3 μg injected per female in 72h-old virgins; *dsOrk*: 2.5–3 μg injected per female in 96h-old virgins; *ds118504077* and *ds118504075*: 3–3.6 μg injected per female in 96h-old virgins. For dual injections of *ds118504077+ds118504075*, 3.5–4.2 μg of a 1:1 mixture of purified dsRNAs was injected per female in 96-hr-old virgins.

To achieve tissue-specific knockdown (abdomen only or head + abdomen) of *sNPF* and *RYa*, injections were performed at different concentrations and on different days. Injecting dsRNA into the abdomens of 3- to 7-hr-old females produced abdomen-specific knockdowns without affecting head expression, whereas injections into head or abdomens of 96-hr-old females resulted in knockdowns in both tissues. Moreover, head knockdowns in older females (96h-old) required higher dsRNA concentrations (3.5–4.2 μg injected per female), with knockdown efficiency correlating with the amount injected. In contrast, abdominal knockdowns in younger females (3- to 7-hr-old) could be achieved even with lower dsRNA amounts (2–4.2 μg injected per female). For dual injections, 3.5–4.2 μg of the 1:1 mixture of purified dsRNAs was injected per female as described above: in 3- to 7-hr-old females for abdomen-specific knockdowns and in 96-hr-old virgins for head + abdomen knockdowns (*Figure 4—figure supplement 2B*).

Injections were performed on a FemtoJet 4i system (Eppendorf) and each female was injected with 0.5–0.6 μL of the respective concentrated dsRNAs in the upper thorax. In some cases, dsRNAs were also injected in the abdomens to ensure that the site of injection does not alter the behaviour. Uninjected or *dsGFP*-injected (same concentration as that of the candidate gene) females were included as a control each time. 15–25 females were injected for each trial and at least two trials from independent batches of mosquitoes were performed for each target gene.

Injected females were allowed to recover in paper cups with *ad libitum* access to sugar and assessed for blood-feeding behaviour at 3 days post-injection, as previously described. Fed and unfed females were visually scored and five to six females were randomly selected for each sample – whole heads and abdomens were collected separately for analysis. Total RNA was extracted using Direct-zol RNA MicroPrep (Zymo Research) columns, as per the manufacturer's protocol. RNA was quantified using Qubit Broad range assay (Invitrogen). For each sample, 200–220 ng RNA was used to prepare cDNA using the SuperScript IV Reverse Transcriptase (Invitrogen 18090050) or Maxima H Minus Reverse transcriptase (Thermo Fisher Scientific EP0752). Quantitative PCR (qPCR) was performed using PowerUp SYBR Green Master Mix (Applied Biosystems) to determine the knockdown efficiency. Relative mRNA abundances were calculated using the delta-delta Ct method (*Livak and Schmittgen, 2001*), normalised to RpS7. Statistical analyses were performed using GraphPad Prism (v10.2.0). All the primers used in the study are listed in *Table 1*.

To assess the sugar-feeding behaviour of dsRNA injected females (*dssNPF*, *dsRYa* and *dsRYa +sNPF*), injections were performed as described above. Injected females were allowed to recover with *ad libitum* access to normal sugar for the first two days, which was then replaced with Rhodamine B-spiked sugar (1 mg/25 ml of sugar, Sigma R6626) for the next 24 hr. Control females were treated similarly. The blood-feeding assay was performed 3 days post-injection as described above, with both blood and sugar provided in parallel. Abdomens were scored visually for ingestion of blood under white light, and under fluorescent microscope (red channel) for ingestion of sugar (*Figure 4—figure supplement 2A*).

## Hybridisation chain reaction (HCR)

Previously published protocol *Bruce et al., 2021* was modified to perform HCR *in situ* hybridisations on brain and gut tissues collected from females under three different conditions: D0 (not interested in blood), D5 virgins (blood-hungry) and mated D2$^{PBM}$(m) (blood-sated). All the reagents, including the probes, amplifiers, and buffers were procured from Molecular Instruments Inc, CA, USA. A set of custom-designed 20 probe pairs were used per target gene: *sNPF (118511038)*, *RYa (118513478)* and their receptors. To find *sNPF* and *RYa* receptors in *An. stephensi*, putative protein sequences were blasted against orthologs in *Drosophila melanogaster*, *Ae. aegypti*, and *An. gambiae*. Based on high identities, consistent in all three organisms, *118517381* and *118507167 (NPY receptor type 2)* were identified as *RYa receptor (RYaR)* and *sNPF receptor (sNPFR)*, respectively.

Five to seven brains for each target, per condition were dissected in ice-cold PBS and fixed with 4% PFA (in PBS) for 20 min at room temperature. Post-washing steps with PTw (0.1% Tween-20 in PBS), four times 10 min each, tissues were permeabilised using detergent solution (50 mM Tris-Cl, pH7.5, 150 mM NaCl, 1 mM EDTA, pH8, 0.5% Tween-20, 1% SDS) for 30 min at room temperature. Brains were then pre-hybridised in probe hybridisation buffer for 30 min at 37 °C, followed by incubation with neuropeptide probes (0.8 pmol each) at 37 °C for 48 hr in a humid chamber. Samples were washed with pre-warmed wash buffer, four times 15 min each, at 37 °C to remove unbound probes. This was followed by two washes, 5 min each with 5 X SSCT (0.1% Tween-20 in 5 X SSC) at room temperature. Samples were then pre-amplified using amplification buffer for 30 min at room temperature. For each sample, 2 μL (3 μM stock) of each hairpins were mixed and prepared by heating at 95 °C for 90 s for annealing. They were allowed to cool to room temperature for 30 min in a dark chamber and 100 μL amplification buffer was added to the mix. Brains were incubated in the hairpin solution for 12–16 hr at 37 °C in dark and excess hairpins were washed off with multiple 5 X SSCT washes: twice for 5 min, twice for 30 min, once for 5 min. For nc82 staining, all the steps were performed in 0.5% PTw buffer (0.5% Tween-20 in PBS). Brains were equilibrated in 0.5% PTw and incubated with anti-mouse nc82 (1:200; DSHB, RRID:AB2314866) for three overnights at 4 °C. Post-washing, goat anti-mouse Alexa-fluor405 (1:400, Invitrogen) was added and samples were left in dark, for one-two overnights at 4 °C. After washing off the unbound secondary antibody, samples

**Table 1.** List of primers used in the study for *dsRNA*-mediated gene knockdown and quantitative PCR (qPCR).

| S. no. | Primer name | Primer sequence |
| --- | --- | --- |
| 1.0 | T7-GFP_F_RNAi | TAATACGACTCACTATAGGGAGACTGGTCGAGCTGGACGGCGA |
| 2.0 | T7-GFP_R_RNAi | TAATACGACTCACTATAGGGAGACTTCTCGTTGGGGTCTTTGCTCAGG |
| 3.0 | T7-RYamide_F_RNAi | TAATACGACTCACTATAGGGATGACCTGGCGAACGATGAAAC |
| 4.0 | T7-RYamide_R_RNAi | TAATACGACTCACTATAGGGGGTCGCCATTCTCGTTGAACTGG |
| 5.0 | RYamide_qPCR_F | ACCGATGCTACAGTCGTGATTCG |
| 6.0 | RYamide_qPCR_R | TAGGCTAGTTGCATGGATCGTACC |
| 7.0 | T7-sNPF_F_RNAi | TAATACGACTCACTATAGGGTTTGATGTCGGAGTCACTGCATCC |
| 8.0 | T7-sNPF_R_RNAi | TAATACGACTCACTATAGGGTCACTCGCATCGTTAACCAACTGC |
| 9.0 | sNPF_qPCR_F | CAATGAGCATCAGCTAGCACCG |
| 10.0 | sNPF_qPCR_R | TCTAGCGCTTTATCCTCGGAGG |
| 11.0 | T7-Hairy_RNAi_F | TAATACGACTCACTATAGGGCTCGTATCAACAACTGTCTGAACG |
| 12.0 | T7-Hairy_RNAi_R | TAATACGACTCACTATAGGGGTACTTGGGTGGTGAGCGTTTGG |
| 13.0 | Hairy_qPCR_F | AATCGTCGGAGCAACAAACCGATC |
| 14.0 | Hairy_qPCR_R | GATGCTTCACCGTCATCTCCAGG |
| 15.0 | T7-Eiger_RNAi_F | TAATACGACTCACTATAGGGCTGGACGATAGCGAGGAAAAGG |
| 16.0 | T7-Eiger_RNAi_R | TAATACGACTCACTATAGGGGCTCATTCGGTTCAGTATTTGTGC |
| 17.0 | Eiger_qPCR_F | GGGCGTAAGACATCACTCACGC |
| 18.0 | Eiger_qPCR_R | TACGATTTCCTTCTGCGTTCCAGG |
| 19.0 | T7-JHEH_RNAi_F | TAATACGACTCACTATAGGGGATTATTGGGGACCCGGAAATGG |
| 20.0 | T7-JHEH_RNAi_R | TAATACGACTCACTATAGGGCGTTTGTGACCGATGCGTTCC |
| 21.0 | JHEH_qPCR_F | ATCGCCAGTGCCAGTTGAGC |
| 22.0 | JHEH_qPCR_R | TTGCTCCATTTCCGGGTCCC |
| 23.0 | T7-Prp_RNAi_F | TAATACGACTCACTATAGGGAATGTATCGCGGTCTGCTTTGC |
| 24.0 | T7-Prp_RNAi_R | TAATACGACTCACTATAGGGCGACTAATTTCCTGCCGTTTGCC |
| 25.0 | Prp_qPCR_F | AAGTTCGGTGCGGCCTGG |
| 26.0 | Prp_qPCR_R | TCTGCTGCTGCTGCTGCTG |
| 27.0 | T7-4075_RNAi_F | TAATACGACTCACTATAGGGGATACGGTCACTTCGCTGGTGG |
| 28.0 | T7-4075_RNAi_R | TAATACGACTCACTATAGGGATGAACAGTACGGTGATGGGCG |
| 29.0 | 4075_qPCR_F | CCTTCGCCTGGATCTATGGCG |
| 30.0 | 4075_qPCR_R | CGAGCGCATTGTAACCGAGTGG |
| 31.0 | T7-4077_RNAi_F | TAATACGACTCACTATAGGGCTCGACACATTCACCTCCATCG |
| 32.0 | T7-4077_RNAi_R | TAATACGACTCACTATAGGGATAAGAATGAGCAGTGTGGCGG |
| 33.0 | 4077_qPCR_F | ATCTACGGCGTGGATCGTATCTGC |
| 34.0 | 4077_qPCR_R | CCTAAAACGTTGTAGCCGATCGGG |
| 35.0 | T7-Orcokinin_F_RNAi | TAATACGACTCACTATAGGGGTCTGTGCCGTACTGCTGTTCG |
| 36.0 | T7-Orcokinin_R_RNAi | TAATACGACTCACTATAGGGGATCGTGCTTCAGGTTGTTACCGG |
| 37.0 | Orcokinin_qPCR_F | GCTGAACGATGGCCAACTAAAGC |
| 38.0 | Orcokinin_qPCR_R | CTTGTCGTACATCGGTGCCAATCC |
| 39.0 | RpS7_qPCR_F | AGGTTGTCGGCAAGCGTATCC |

*Table 1 continued on next page*

Table 1 continued

| S. no. | Primer name | Primer sequence |
| --- | --- | --- |
| 40.0 | RpS7_qPCR_R | AGCTTCTTGTACACCGACGCG |

were equilibrated for at least an hour in Vectashield and mounted in Vectashield for imaging. *In situ* hybridisations against neuropeptide receptors were performed the same way, except the washing steps to remove the unbound probes were performed at 42 °C and samples were incubated with the hairpin solution at room temperature (to minimise background). As controls for background signal, samples were prepared the same way, except no probes were added in the hybridisation buffer (*Figure 4—figure supplement 3E*). Samples for all three conditions were dissected, processed and imaged the same day.

Protocol for gut *in situs* was adapted from *Slaidina et al., 2020*. Five to eight gut samples for each target, per condition were dissected in ice-cold PBS and fixed with 4% PFA (in PTw) for 20 min at room temperature. Post-washing steps with PTw, tissues were dehydrated on ice with graded methanol washes: 25% methanol (in PTw), 50% methanol (in PTw), 75% methanol (in PTw), and 100% methanol for 10 min each. Samples were stored in 100% methanol for up to a month, at –20 °C. For rehydration, samples were washed sequentially with graded methanol: 75% methanol (in PTw), 50% methanol (in PTw), 25% methanol (in PTw), followed by a wash in permeabilisation solution (1% Triton-X in PBS). Samples were permeabilised for 2 hr at room temperature and fixed again with 4% PFA (in PTw) for 20 min. Following 10 min washes: washed once with PTw, twice with 50% PTw/50% 5 X SSCT and twice with 5 X SSCT, samples were pre-hybridised in probe hybridisation buffer for 30 min at 37 °C, followed by incubation with probes (0.8 pmol each) at 37 °C for 48 hr in a humid chamber. Post washing steps with wash buffer at 37 °C and 5 X SSCT at room temperature, samples were pre-amplified in amplification buffer for 30 min at room temperature. Hairpins were prepared the same way as described above and samples were incubated in hairpin solutions for 12–16 hr in dark at room temperature. Following two washes 5 min each with 5 X SSCT, samples were incubated with DAPI (1:1000 in 5 X SSCT) for 1 hr at room temperature. Samples were washed with 5 X SSCT, four times, 10 min each and equilibrated in Vectashield for at least an hour before mounting. As controls for background signal, samples were prepared the same way, except no probes were added in the hybridisation buffer (*Figure 4—figure supplement 4G and H*). Samples for different conditions, prepared on different days and stored at –20 °C, were processed and imaged together. Statistical analyses were performed using GraphPad Prism (v10.2.0).

All samples were imaged at Central Imaging and Flow Cytometry Facility (CIFF) at the Bangalore Life Science Cluster (BLiSC), India. Images were acquired with a 40 X/1.3NA or 10 X/0.4NA (whole guts) objectives at a resolution of 512X512 pixels, on an Olympus FV3000 system. Images were viewed and manually analysed using ImageJ/FIJI (*Schindelin et al., 2012*). Maximum intensity projections of all z-stacks in an image are shown, except in *Figure 4I*, *Figure 4—figure supplement 3B, D*, where stacks were manually selected to capture relevant brain structures.

## Acknowledgements

This work was supported by TIGS. We wish to acknowledge the insectary and its staff at TIGS. All sequencing was done at the Next Generation Genomics Facility (NGGF) at the Bangalore Life Science Cluster. All imaging was done at the Central Imaging and Flow Cytometry Facility (CIFF) at the Bangalore Life Science Cluster. We are grateful to Nitin Gupta, Mahul Chakraborty, Tussar Saha, Sriram Narayanan, Farah Ishtiaq, Kavita Isvaran, Sunil Laxman, Chris Q Doe, Claude Desplan, Krishna Melnattur, Suresh Subramani, K VijayRaghavan, the Indian Neurobehaviour group, and the Vosshall lab for their invaluable input and discussion during the course of this project and comments on the manuscript.

# Additional information

### Competing interests
Sonia Q Sen: Senior editor, eLife. The other authors declare that no competing interests exist.

### Funding

| Funder | Grant reference number | Author |
| --- | --- | --- |
| Tata Institute for Genetics and Society | | Sonia Q Sen |

The funders had no role in study design, data collection and interpretation, or the decision to submit the work for publication.

### Author contributions
Prashali Bansal, Conceptualization, Data curation, Formal analysis, Validation, Investigation, Visualization, Methodology, Writing – original draft, Writing – review and editing; Roshni Pillai, Pooja DB, Data curation, Formal analysis, Writing – review and editing; Sonia Q Sen, Conceptualization, Resources, Supervision, Funding acquisition, Investigation, Writing – original draft, Project administration, Writing – review and editing

### Author ORCIDs
Prashali Bansal (iD) https://orcid.org/0000-0003-3288-9855
Roshni Pillai (iD) https://orcid.org/0009-0005-6299-6514
Sonia Q Sen (iD) https://orcid.org/0000-0003-4693-3378

### Ethics
The experiments involving human blood were approved by the Institutional Human Ethics Committee: inStem/IEC-22/02.

Reviewer #2 (Public review): https://doi.org/10.7554/eLife.108625.4.sa1
Author response https://doi.org/10.7554/eLife.108625.4.sa2

---

# Additional files

### Supplementary files
MDAR checklist

Supplementary file 1. Summary of statistical analysis of feeding behaviours using Generalised Linear Mixed Models (GLMM).

Supplementary file 2. Table of transcript abundance (in TPM values) for genes identified in the central brain RNAseq data across the gonotrophic cycle.

Supplementary file 3. Table of differentially expressed genes (DEGs) identified across states of blood-hunger and -satiety.

### Data availability
Raw fastq files have been deposited at the National Center for Biotechnology Information (NCBI) Sequence Read Archive (SRA), under the BioProject ID PRJNA1297931.

The following dataset was generated:

| Author(s) | Year | Dataset title | Dataset URL | Database and Identifier |
| --- | --- | --- | --- | --- |
| Bansal P, Pillai R, Babu PD, Sen SQ | 2025 | Two neuropeptides that promote blood-feeding in Anopheles stephensi mosquitoes | https://www.ncbi.nlm.nih.gov/bioproject/PRJNA1297931 | NCBI BioProject, PRJNA1297931 |

The following previously published datasets were used:

| Author(s) | Year | Dataset title | Dataset URL | Database and Identifier |
| --- | --- | --- | --- | --- |
| Prasad TSK, Mohanty AK | 2017 | Integrating transcriptomics and proteomics data for accurate assembly and annotation of genomes | https://www.ncbi.nlm.nih.gov/sra/SRX620223 | NCBI Sequence Read Archive, SRX620223 |
| Prasad TSK, Mohanty AK | 2017 | Integrating transcriptomics and proteomics data for accurate assembly and annotation of genomes | https://www.ncbi.nlm.nih.gov/sra/SRX618937 | NCBI Sequence Read Archive, SRX618937 |
| Prasad TSK, Mohanty AK | 2017 | Integrating transcriptomics and proteomics data for accurate assembly and annotation of genomes | https://www.ncbi.nlm.nih.gov/sra/SRX620224 | NCBI Sequence Read Archive, SRX620224 |

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
