## [Editor Report · eLife Assessment]

This is a **valuable** study that integrates behavioral and molecular approaches to identify neuromodulators influencing blood-feeding behavior in the disease vector *Anopheles stephensi*. Through gene expression analyses across blood-seeking life stages and RNA interference experiments, the authors present **solid** evidence that co-knockdown of the neuromodulators *short neuropeptide F* and *RYamide* affects blood-seeking states in *A. stephensi*. However, evidence demonstrating that these neuropeptides are sufficient to promote host-seeking is lacking.

---

## [Referee Report · Reviewer #2 (Public review)]

Summary:

In this study, Bansal et al examine and characterize feeding behaviour in Anopheles stephensi mosquitoes. While sharing some similarities to the well-studied Aedes aegypti mosquito, the authors demonstrate that mated-females, but not unmated (virgin) females, exhibit suppression in their blood-feeding behaviour after imbibing an initial bloodmeal. Using brain transcriptomic analysis comparing sugar fed, blood fed and starved mosquitoes, several candidate genes potentially responsible for influencing blood-feeding behaviour were identified, including two neuropeptides (short NPF and RYamide) that are known to modulate feeding behaviour in other mosquito species. Using molecular tools including *in situ* hybridization, the authors map the distribution of cells producing these neuropeptides in the nervous system and in the gut. Further, by implementing systemic RNA interference (RNAi), the study suggests that both neuropeptides (particularly in the brain, but not in the abdomen since knockdown outside the brain did not affect feeding behaviour) appear to promote blood-feeding while having no impact on sugar feeding. Interestingly, when either of these two neuropeptide gene transcripts were reduced independently by RNAi, the proportion of females acquiring a blood meal was not affected, whereas simultaneous knockdown of both sNPF and RYa led to a reduction in blood feeding behaviour but did not impact sugar feeding.

Given that the expression of both neuropeptide genes was found in mostly in non-overlapping brain neurons, this suggests that these two neuropeptides may elicit at least partially complementary actions promoting blood feeding in A. stephensi. Indeed, their putative receptors appear to be colocalized within several neurons within the brain, which could explain why knockdown of both sNPF and RYa transcripts was required to affect blood feeding behaviour (although authors could not confirm if either of these neuropeptides act independently as only partial knockdown was achieved in the brain). Finally, while sNPF was mapped to brain neurons and midgut enteroendocrine cells, the authors mapped RYa only in the brain while reporting expression in the abdomen by qPCR, but that was not localized to the midgut EECs (like sNPF). Therefore, the source of RYamide in the abdomen remains unknown in this mosquito species, but could involve the abdominal ganglia where this neuropeptide has been localized in Ae. aegypti.

Strengths and/or weaknesses:

Overall, the manuscript was effectively communicated. Previous concerns and requested clarifications have been addressed in the revised manuscript. While advanced cell-specific tools are lacking in this mosquito species, one weakness here is that peptides could have been applied ectopically in attempts to rescue the deficit in blood feeding behaviour following knockdown by RNAi. Further insight in this regard may be provided in future studies by this and other research groups.

Reviewing editor comment:

Inclusion of a schematic in Supplementary Figure S9B addresses the point raised by reviewer 1 in the previous round.

---

## [Author Response]

The following is the authors’ response to the previous reviews

**Public Reviews:**

**Reviewer #1 (Public review):**
Summary:Here Bansal et al., present a study on the fundamental blood and nectar feeding behaviors of the critical disease vector, Anopheles stephensi. The study encompasses not just the fundamental changes in blood feeding behaviors of the crucially understudied vector, but then use a transcriptomic approach to identify candidate neuromodulation path ways which influence blood feeding behavior in this mosquito species. The authors then provide evidence through RNAi knockdown of candidate pathways that the neuromodulators sNPF and Rya modulate feeding either via their physiological activity in the brain alone or through joint physiological activity along the brain-gut axis (but critically not the gut alone). Overall, I found this study to be built on tractable, well-designed behavioral experiments.Their study begins with a well-structured experiment to assess how the feeding behaviors of A. stephensi changes over the course of its life history and in response to its age, mating and oviposition status. The authors are careful and validate their experimental paradigm in the more well-studied Ae. aegypti, and are able to recapitulate the results of prior studies which show that mating is pre-requisite for blood feeding behaviors in Ae. aegypt. Here they find A. stephensi like another Anopheline mosquitoes has a more nuanced regulation of its blood and nectar feeding behaviors.The authors then go on to show in a Y- maze olfactometer that to some degree, changes in blood feeding status depend on behavioral modulation to host-cues, and this is not likely to be a simple change to the biting behaviors alone. I was especially struck by the swap in valence of the host-cues for the blood-fed and mated individuals which had not yet oviposited. This indicates that there is a change in behavior that is not simply desensitization to host-cues while navigating in flight, but something much more exciting happening.The authors then use a transcriptomic approach to identify candidate genes in the blood feeding stages of the mosquito's life cycle to identify a list of 9 candidates which have a role in regulating the host-seeking status of A. stephensi. Then through investigations of gene knockdown of candidates they identify the dual action of RYa and sNPF and candidate neuromodulators of host-seeking in this species. Overrall, I found the experiments to be welldesigned. I found the molecular approach to be sound. While I do not think the molecular approach is necessarily an all-encompassing mechanism identification (owing mostly to the fact that genetic resources are not yet available in A. stephensi as they are in other dipteran models), I think it sets up a rich lines of research questions for the neurobiology of mosquito behavioral plasticity and comparative evolution of neuromodulator action.Strengths:I am especially impressed by the authors' attention to small details in the course of this article. As I read and evaluated this article I continued to think how many crucial details I may have missed if I were the scientist conducting these experiments. That attention to detail paid off in spades and allowed the authors to carefully tease apart molecular candidates of blood-seeking stages. The authors top down approach to identifying RYamide and sNPF starting from first principles behavioral experiments is especially comprehensive. The results from both the behavioral and molecular target studies will have broad implications for the vectorial capacity of this species and comparative evolution of neural circuit modulation.I believe the authors have adequately addressed all of my concerns; however, I think an accompanying figure to match the explained methods of the tissue-specific knockdown would help readers. The methods are now explicitly written for the timing and concentrations required to achieve tissue-specific knockdown, but seeing the data as a supplement would be especially reassuring given the critical nature of tissue-specific knockdown to the final interpretations of this paper.

We thank the reviewer for the suggestion and have now incorporated a schematic in the supplementary figure S9B, explaining our methodology for achieving tissue-specific knockdowns.

**Reviewer #2 (Public review):**
Summary:In this manuscript, Bansal et al examine and characterize feeding behaviour in Anopheles stephensi mosquitoes. While sharing some similarities to the well-studied Aedes aegypti mosquito, the authors demonstrate that mated-females, but not unmated (virgin) females, exhibit suppression in their blood-feeding behaviour. Using brain transcriptomic analysis comparing sugar fed, blood fed and starved mosquitoes, several candidate genes potentially responsible for influencing blood-feeding behaviour were identified, including two neuropeptides (short NPF and RYamide) that are known to modulate feeding behaviour in other mosquito species. Using molecular tools including in situ hybridization, the authors map the distribution of cells producing these neuropeptides in the nervous system and in the gut. Further, by implementing systemic RNA interference (RNAi), the study suggests that both neuropeptides appear to promote blood-feeding (but do not impact sugar feeding) although the impact was observed only after both neuropeptide genes underwent knockdown.While the authors have addressed most of the concerns of the original manuscript, a few issues remain. Particularly, the following two points:(5) Figure 4The authors state that there is more efficient knockdown in the head of unfed females; however, this is not accurate since they only get knockdown in unfed animals, and no evidence of any knockdown in fed animals (panel D). This point should be revised in the results test as well.Perhaps we do not understand the reviewer's point or there has been a misunderstanding. In Figure 4D, we show that while there is more robust gene knockdown in unfed females, bloodfed females also showed modest but measurable knockdowns ranging from 5-40% for RYamide and 2-21% for sNPF.NEW-In both the dsRNA treatments where animals were fed, neither was significantly different from control. Therefore, there is no change, and indeed this is confirmed by the author's labelling of the figure stats in panel 4D.

We agree with the reviewer and thank them for pointing it out. We have now revised the figure legend and the text to reflect these results (see **lines 351-354**).

In addition, do the uninjected and dsGFP-injected relative mRNA expression data reflect combined RYa and sNPF levels? Why is there no variation in these data,...In these qPCRs, we calculated relative mRNA expression using the delta-delta Ct method (see line 975). For each neuropeptide its respective control was used. For simplicity, we combined the RYa and sNPF control data into a single representation. The value of this control is invariant because this method sets the control baseline to a value of 1.NEW-The authors are claiming that there is no variation between individual qPCR experiments (particularly in their controls)? Normally, one uses a known standard value (or calibrator) across multiple experiments/plates so that variation across biological replicates can be assessed. This has an impact on statistical analyses since there is no variation in the control data. Indeed, this impacts all figures/datasets in the manuscript where qPCR data is presented. All the controls have zero variation!

We are truly thankful to this reviewer for insisting on this point. It has made us revisit what we thought we understood and now realise were doing wrong (though many in literature do it this way!). We were – incorrectly – setting each control to 1 and calculating relative fold changes for each replicate independently. While this is often seen in literature, we now realise that it is incorrect. We have revisited all our analyses and normalized all samples to the mean ΔCt of the control group, which captures biological variation in both control and experimental groups. All data are now re-plotted to show individual data points for both control and experimental groups, and the error bars on controls represent the biological variation across replicates (Figure 4D, 4F, 4G, S8, S9). Statistical analyses were also revised accordingly, and, importantly, they do not change any conclusions. Please note that the abdominal expression of *sNPF* and *RYa* are so low that the controls show very variable baseline expression values.

**Reviewer #3 (Public review):**
Summary:This manuscript investigates the regulation of host-seeking behavior in Anopheles stephensi females across different life stages and mating states. Through transcriptomic profiling, the authors identify differential gene expression between "blood-hungry" and "blood-sated" states. Two neuropeptides, sNPF and RYamide, are highlighted as potential mediators of host-seeking behavior. RNAi knockdown of these peptides alters host-seeking activity, and their expression is anatomically mapped in the mosquito brain (sNPF and RYamide) and midgut (sNPF only).Strengths:(1) The study addresses an important question in mosquito biology, with relevance to vector control and disease transmission.Transcriptomic profiling is used to uncover gene expression changes linked to behavioral states.(2) The identification of sNPF and RYamide as candidate regulators provides a clear focus for downstream mechanistic work.(3) RNAi experiments demonstrate that these neuropeptides are necessary for normal hostseeking behavior.(4) Anatomical localization of neuropeptide expression adds depth to the functional findings.Weaknesses:(1) The title implies that the neuropeptides promote host-seeking, but sufficiency is not demonstrated and some conclusions appear premature based on the current data. The support for this conclusion would be strengthened with functional validation using peptide injection or genetic manipulation.(2) The identification of candidate receptors is promising, but the manuscript would be significantly strengthened by testing whether receptor knockdowns phenocopy peptide knockdowns. Without this, it is difficult to conclude that the identified receptors mediate the behavioral effects.(3) Some important caveats, such as variation in knockdown efficiency and the possibility of offtarget effects, are not adequately discussed.

These comments were addressed in the previous round.

**Recommendations for the authors:**

**Reviewer #1 (Recommendations for the authors):**
Awesome paper everyone. A delight to read and review.

Thank you very much! We appreciated your comments too!